# Genomic deletion of Bcl6 differentially affects conventional dendritic cell subsets and compromises Tfh/Tfr/Th17 cell responses

Hongkui Xiao [1,13], Isabel Ulmert [1,13], Luisa Bach[2], Johanna Huber[3], Hamsa Narasimhan[4,5], Ilia Kurochkin[6,7], Yinshui Chang [2,3], Signe Holst [1,8,9], Urs Mörbe [1], Lili Zhang[10], Andreas Schlitzer[10], Carlos-Filipe Pereira [6,7,11], Barbara U. Schraml [4,5], Dirk Baumjohann [2,3] ✉ & Katharina Lahl [1,8,9,12] ✉

Conventional dendritic cells (cDC) play key roles in immune induction, but what drives their heterogeneity and functional specialization is still ill-defined. Here we show that cDC-specific deletion of the transcriptional repressor Bcl6 in mice alters the phenotype and transcriptome of cDC1 and cDC2, while their lineage identity is preserved. Bcl6-deficient cDC1 are diminished in the periphery but maintain their ability to cross-present antigen to CD8⁺ T cells, confirming general maintenance of this subset. Surprisingly, the absence of Bcl6 in cDC causes a complete loss of Notch2-dependent cDC2 in the spleen and intestinal lamina propria. DC-targeted Bcl6-deficient mice induced fewer T follicular helper cells despite a profound impact on T follicular regulatory cells in response to immunization and mounted diminished Th17 immunity to *Citrobacter rodentium* in the colon. Our findings establish Bcl6 as an essential transcription factor for subsets of cDC and add to our understanding of the transcriptional landscape underlying cDC heterogeneity.

Conventional dendritic cells (cDC) have the unique ability to present antigens to naïve T cells within lymphoid organs[1–3]. Upon encounter of antigen, migratory cDC scanning the peripheral environment mature and relocate to the draining lymph nodes (LN), where they collaborate with LN-resident cDC to initiate immunity[4]. The spleen is the secondary lymphoid organ responsible for the induction of immune responses to blood-derived antigen[5], and maturation of splenic cDC leads to similar relocation from the marginal zone and red pulp into the white pulp,

¹Section for Experimental and Translational Immunology, Institute for Health Technology, Technical University of Denmark (DTU), 2800 Kongens, Lyngby, Denmark. ²Medical Clinic III for Oncology, Hematology, Immuno-Oncology and Rheumatology, University Hospital Bonn, University of Bonn, Bonn, Germany. ³Institute for Immunology, Biomedical Center, Faculty of Medicine, LMU Munich, Planegg-Martinsried, Munich, Germany. ⁴Biomedical Center, Institute of Cardiovascular Physiology and Pathophysiology, Faculty of Medicine, Ludwig-Maximillians-Universität München, Planegg-Martinsried, Munich, Germany. ⁵Walter-Brendel-Centre of Experimental Medicine, University Hospital, LMU Munich, Planegg-Martinsried, Munich, Germany. ⁶Cell Reprogramming in Hematopoiesis and Immunity Laboratory, Lund Stem Cell Center, Molecular Medicine and Gene Therapy, Lund University, Lund, Sweden. ⁷Wallenberg Center for Molecular Medicine, Lund University, Lund, Sweden. ⁸Department of Microbiology, Immunology, and Infectious Diseases, Cumming School of Medicine, University of Calgary, Calgary, Alberta, Canada. ⁹Calvin, Phoebe and Joan Snyder Institute for Chronic Diseases, University of Calgary, Calgary, Alberta, Canada. ¹⁰Quantitative Systems Biology, Life and Medical Sciences (LIMES) Institute, University of Bonn, Bonn, Germany. ¹¹Center for Neuroscience and Cell Biology, University of Coimbra, Coimbra, Portugal. ¹²Immunology Section, Lund University, Lund 221 84, Sweden. ¹³These authors contributed equally: Hongkui Xiao, Isabel Ulmert. ✉e-mail: dirk.baumjohann@uni-bonn.de; katharina.lahl@ucalgary.ca

where naïve B and T cells reside[6,7]. "Resident" resting DC are also found in the spleen, where they are located in the T cell zones in the absence of stimulation[8].

cDC are broadly divided into two main subsets, cDC1 and cDC2[1]. cDC1 are characterized by their expression of the cDC1-exclusive chemokine receptor XCR1 and the absence of CD11b and Sirpα. Their development and functions are dependent on the transcription factors (TF) IRF8[9,10], BATF3[11], Nfil3[12,13], and Id2[14], and they excel at cross-presenting antigen to CD8[+] T cells, which is at least in part attributable to their ability to engulf apoptotic material[15–19]. cDC2 express high levels of CD11b and Sirpα, but not XCR1 and the transcriptional network governing their development is less well defined. cDC2 show a higher level of heterogeneity than cDC1[20], and recent attempts to subset cDC2 further have suggested separation based on Notch2 versus Klf4-dependency[21]. IRF4-deficiency leads to a partial loss of cDC2 and is thought to impact primarily on their survival and ability to migrate[22–24]. Other TFs reported to influence cDC2 development or function include ZEB2[25], IRF2[26], and Rel-B[27]. In the spleen, the majority of cDC2 express endothelial cell-specific adhesion molecule (ESAM), a marker defining cDC2 located in the bridging channels of the marginal zone, where they fulfill critical functions in activating CD4[+] T cells and germinal center (GC) formation[28,29]. The development of ESAM[hi] cDC2 depends on Notch2-signaling induced by delta-like ligand 1 (DLL1) expressed by stromal cells[22,28,30,31] and on the chemokine receptor Ebi2 (GPR183)[32].

The TF Bcl6 plays an essential role in GC reactions, being required for the differentiation of both germinal center (GC) B cells and T follicular helper (Tfh) cells in a cell-intrinsic manner[33]. Bcl6 is expressed in cDC1 and to a lower extent in cDC2[20], and we have shown previously that complete Bcl6-deficiency causes a loss of CD11b[−] cDC, presumed to be cDC1[20]. Other work has confirmed that Bcl6-deficient mice lack splenic CD8-expressing cDC1 but also reported a reduction in splenic CD4[+] cDC2[34]. However, a more recent study has disputed these findings by showing normal cDC1 numbers in mice with conditional *Bcl6* deletion driven by cre recombinase under the control of the *Csf1* promoter. Importantly, these authors suggested that the downregulation of CD11c in the absence of Bcl6 may have compromised the detection of cDC1 in the earlier studies[35].

Here we reconcile these conflicting data by performing in-depth analysis of the cDC compartment in mice lacking Bcl6 in all cDC or specifically in cDC1, using CD11c- and Clec9a- or XCR1-driven cre drivers, respectively. Our results show that while cDC1 showed substantial alterations in gene expression including deregulation of genes involved in cytokine signaling and cell adhesion, their ability to cross-present antigen is largely conserved in the absence of Bcl6. Bcl6-deficiency in DC leads to a complete loss of the Notch2-dependent ESAM[hi] cDC2 subset in the spleen and consequently to alterations in splenic Tfh and T follicular regulatory (Tfr) cells as well as antibody responses following immunization. Likewise, Notch2-dependent cDC2 in the intestinal lamina propria (LP) are reduced in the absence of Bcl6 in DC, leading to a decrease in T helper 17 cell (Th17)-responses and delayed healing upon infection with the enteric pathogen *Citrobacter rodentium*. Our findings suggest that the core subset identities of cDC1 and cDC2 are maintained in the absence of Bcl6 but that its deficiency in DC causes cell-intrinsic, subset-specific alterations in DC abundance and function, which translates into corresponding defects in adaptive immune responses.

## Results
### Bcl6-deficiency alters DC subset phenotypes and numbers across tissues
We have previously shown that mice completely lacking Bcl6 are largely devoid of CD11b[−] cDC, presumed to be cDC1[20]. To better define the impact of Bcl6-deficiency on cDC1, we generated mice lacking Bcl6, specifically in cDC1, by crossing XCR1.cre mice to mice carrying a

floxed Bcl6 gene (*XCR1.Bcl6[KO]*). Additionally, we deleted Bcl6 in all CD11c-expressing cells by crossing CD11c.cre mice to Bcl6[flox] mice (*CD11c.Bcl6[KO]*). As controls, we used the respective cre negative Bcl6[flox] littermates (designated as control throughout). As expected, flow cytometry revealed mutually exclusive populations of CD11c[+]MHCII[+] cDC in control spleens based on the expression of XCR1 or CD11b to identify cDC1 and cDC2, respectively. However, there was a clear population of XCR1[+]CD11b[+] cDC in the absence of Bcl6 in the spleens of both Bcl6-deficient mouse lines (Fig. 1A, for complete gating strategies see Suppl. Fig. 1). XCR1[+]CD11b[+] cDC were also abundant within the resident populations of peripheral and mesenteric lymph node (pLN and mLN) cDC in the absence of Bcl6 (Fig. 1B). The absolute numbers of total XCR1[+] cDC (independent of CD11b expression) were only mildly reduced in *XCR1.Bcl6[KO]* and normal in *CD11c.Bcl6[KO]* mice, while XCR1[−]CD11b[+] cDC were significantly diminished in *CD11c.Bcl6[KO]* mice, but normal in *XCR1.Bcl6[KO]* mice (Fig. 1A). Resident cDC numbers were not substantially altered in either pLNs or mLNs (Fig. 1B).

We next analyzed migratory DC populations in different organs. Mice lacking Bcl6 in cDC1 or in all DC had significantly fewer migratory XCR1[+] cDC in pLNs and mLNs, and these also expressed CD11b (Fig. 2A, for complete gating strategies, see Suppl. Fig. 1). Interestingly, we also detected a loss of XCR1[−]CD11b[+]CD103[+] cDC in the migratory compartment of the mLNs of *CD11c.Bcl6*[KO] mice, whereas the numbers of XCR1[−]CD11b[+]CD103[−] cDC were normal (Fig. 2A). As the majority of migratory cDC in the mLNs derive from the small intestinal lamina propria (SILP), we next analyzed this compartment. There were fewer XCR1[+] cDC in the SILP of both *XCR1.Bcl6[KO]* and *CD11c.Bcl6[KO]* mice, all of which again expressed CD11b (Fig. 2B). In addition, the numbers of XCR1[−]CD11b[+]CD103[+] cDC were reduced in *CD11c.Bcl6[KO]* mice, while the numbers of XCR1[−]CD11b[+]CD103[−] cDC were less affected (Fig. 2B). Together, these results suggest that the decrease in migratory XCR1[+] and XCR1[−]CD11b[+]CD103[+] cDC seen in mLNs are caused by the absence of these populations, rather than a disability to migrate. Similar changes in cDC were seen in the lung and the blood in the absence of Bcl6, with reduced numbers and upregulation of CD11b in XCR1[+] cDC in both *XCR1.Bcl6[KO]* and *CD11c.Bcl6[KO]* mice, together with reduced numbers of XCR1[−]CD11b[+] cDC in *CD11c.Bcl6[KO]* mice (Suppl. Fig. 2). Interestingly, while XCR1[+] DCs upregulated CD11b in all organs, XCR1[−]CD11b[+] DCs further upregulated CD11b particularly in systemic populations (spleen and resident DCs from the mLN and pLN), while peripheral XCR1[−]CD11b[+] DC showed no further increase of their naturally high levels of CD11b (Suppl. Fig. 3). Thus, Bcl6 deficiency affects cDC in several locations.

### cDC1 and cDC2 signatures are preserved in the absence of Bcl6 in DC
One of the most surprising findings from our studies was the expression of the cDC2 marker CD11b by the XCR1[+] cDC that were preserved in Bcl6-deficient mice. To determine whether this reflects the acquisition of CD11b by cDC1 or cDC2 with aberrant XCR1 expression, we sorted splenic XCR1[+] cDC (independent of CD11b expression) and XCR1[−]CD11b[+] cDC from control, *XCR1.Bcl6[KO]* and *CD11c.Bcl6[KO]* mice for bulk RNA-sequencing (see Fig. 1A for gating). Principal component (PC) analysis showed that XCR1[+] cDC segregated from XCR1[−]CD11b[+] cDC in all mouse models along PC axis 1, which accounted for 69% of the variance. This strongly supports the idea that Bcl6-deficient cDC1 and cDC2 maintained overall lineage identity (Fig. 3A). Nevertheless, PC2 separated the populations on the basis of Bcl6 presence or absence, accounting for 13% of the variance and suggesting that Bcl6 has a significant impact on the gene transcription profile of both cDC1 and cDC2. Interestingly, cDC1 from *XCR1.Bcl6[KO]* and *CD11c.Bcl6[KO]* mice clustered closely and were separated from control cDC1, while cDC2 in *CD11c.Bcl6[KO]* mice were segregated from those in *XCR1.Bcl6[KO]* and control mice, consistent with XCR1-cre being specific to cDC1 and indicating that Bcl6 regulates gene expression in both cDC subsets in a

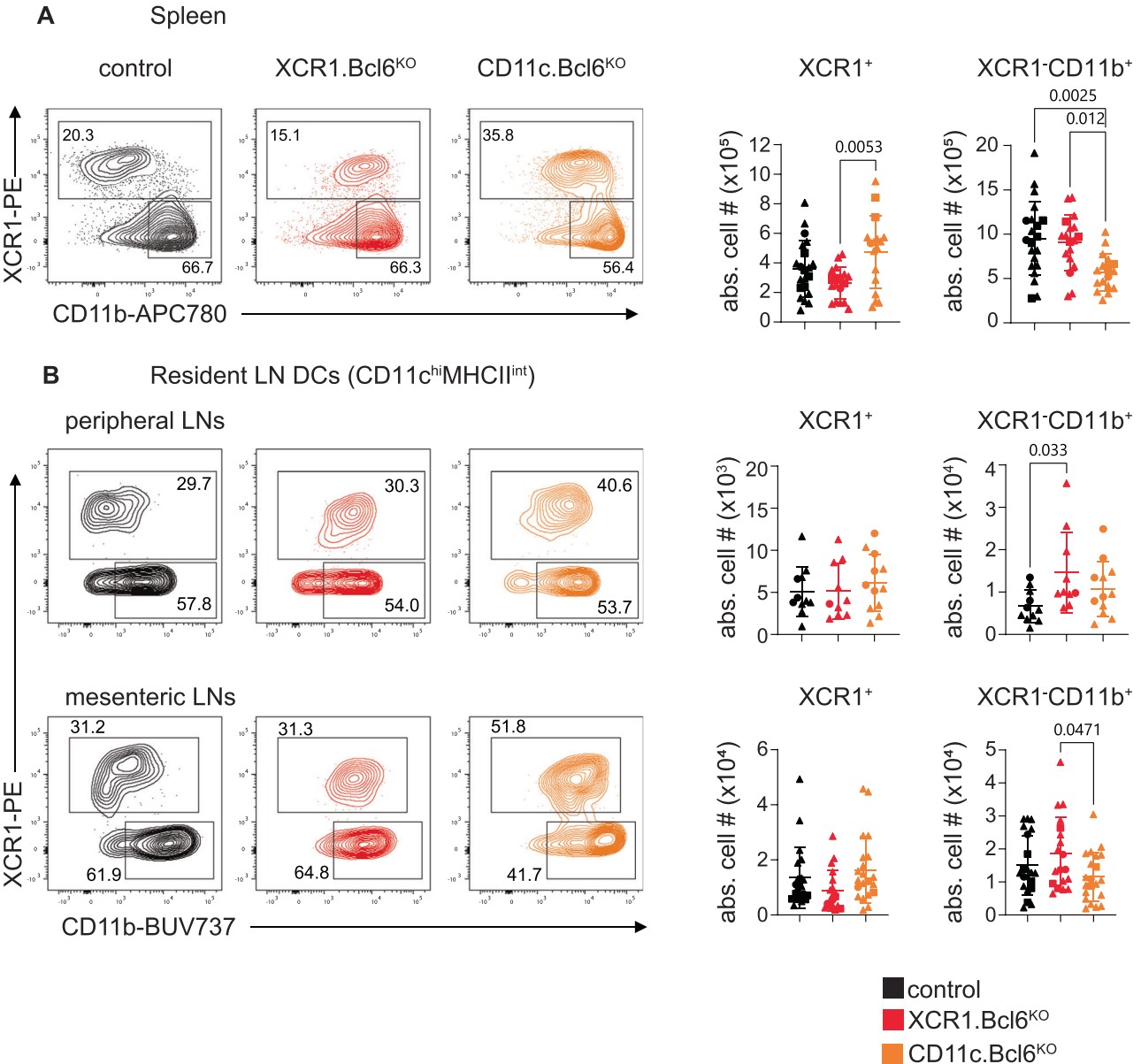

**Fig. 1 | Phenotype and abundance of lymphoid tissue-resident cDC lacking Bcl6. A** Left: Splenic XCR1+ and XCR1−CD11b+ dendritic cell (DC) subsets in control (black), *XCR1.Bcl6*^KO (red), and *CD11c.Bcl6*^KO (orange) mice. Plots show representative staining profiles of gated live, single lineage− (CD3, CD19, CD64, B220, NK1.1), MHCII+, and CD11c^hi splenocytes. Right: Absolute numbers of splenic XCR1+ and XCR1−CD11b+ DC from control, *XCR1.Bcl6*^KO, and *CD11c.Bcl6*^KO mice. Statistical analysis by one-way ANOVA, exact *P* values are annotated. Data points represent values from individual mice pooled from 8 to 9 experiments with 1–5 mice per group, and lines represent means ± SD. **B** Left: Flow cytometry plots showing resident XCR1+ and XCR1−CD11b+ DC subsets in the peripheral and mesenteric lymph nodes (pLN and mLN) of control, *XCR1.Bcl6*^KO, and *CD11c.Bcl6*^KO mice. Plots show representative staining profiles of gated single live, CD45+, lineage− (CD3, CD19, CD64, B220, NK1.1), CD11c^hiMHCII^int resident DC. Right: Absolute numbers of resident cDC subsets in the pLN and mLN of control, *XCR1.Bcl6*^KO, and *CD11c.Bcl6*^KO mice. Statistical analysis by one-way ANOVA, exact *P* values are annotated. Data points represent values from individual mice pooled from 4 (pLN) and 8 (mLN) experiments with 2–3 (pLN) and 1–6 mice (mLN) per group, and lines represent means ± SD. Circles: females; triangles: males; squares: sex-information missing.

cell-intrinsic manner. PC3 accounted for 4% of the variance and showed opposite directional bias in Bcl6-deficient cDC1 and cDC2 (Fig. 3A). This suggests that Bcl6-mediated regulation of gene transcription may affect some targets in cDC1 and cDC2 differently.

To explore further how Bcl6 might regulate cDC lineage identity, we generated cDC1 and cDC2 signatures by identifying the top 50 genes that were differentially expressed between our control cDC1 and cDC2 and interrogated how these signatures were affected by a loss of Bcl6 expression (for gene lists and expression levels see Suppl. Data 1). Supporting their maintenance of lineage identity, XCR1+ cDC from both *XCR1.Bcl6*^KO and *CD11c.Bcl6*^KO mice retained the expression of

cDC1 markers, including *Irf8*, *Tlr3*, *Clec9a*, and *Cadm1* (Fig. 3B). However, *Cd8a* was downregulated in Bcl6-deficient XCR1+ cDC and several genes associated with cDC2 identity were upregulated by XCR1+ cDC in the absence of Bcl6, including *Sirpa*, *Zeb2*, *Csf1r*, and *Itgam* (encoding CD11b) (foldchange > 2, padjust <0.01). Although Bcl6-deficient CD11b+ cDC2 did not express cDC1-specific markers, cDC2 from *CD11c.Bcl6*^KO mice showed altered expression of several classical cDC2 markers, of which some were further upregulated, such as *Csf1*, *Zeb2*, and *Itgam*, while others were downregulated, including *Clec4a4* and *CD4* (Fig. 3B). Despite maintenance of core lineage expression patterns, the overall transcriptional profiles of XCR1+ and XCR1−CD11b+ cDC were

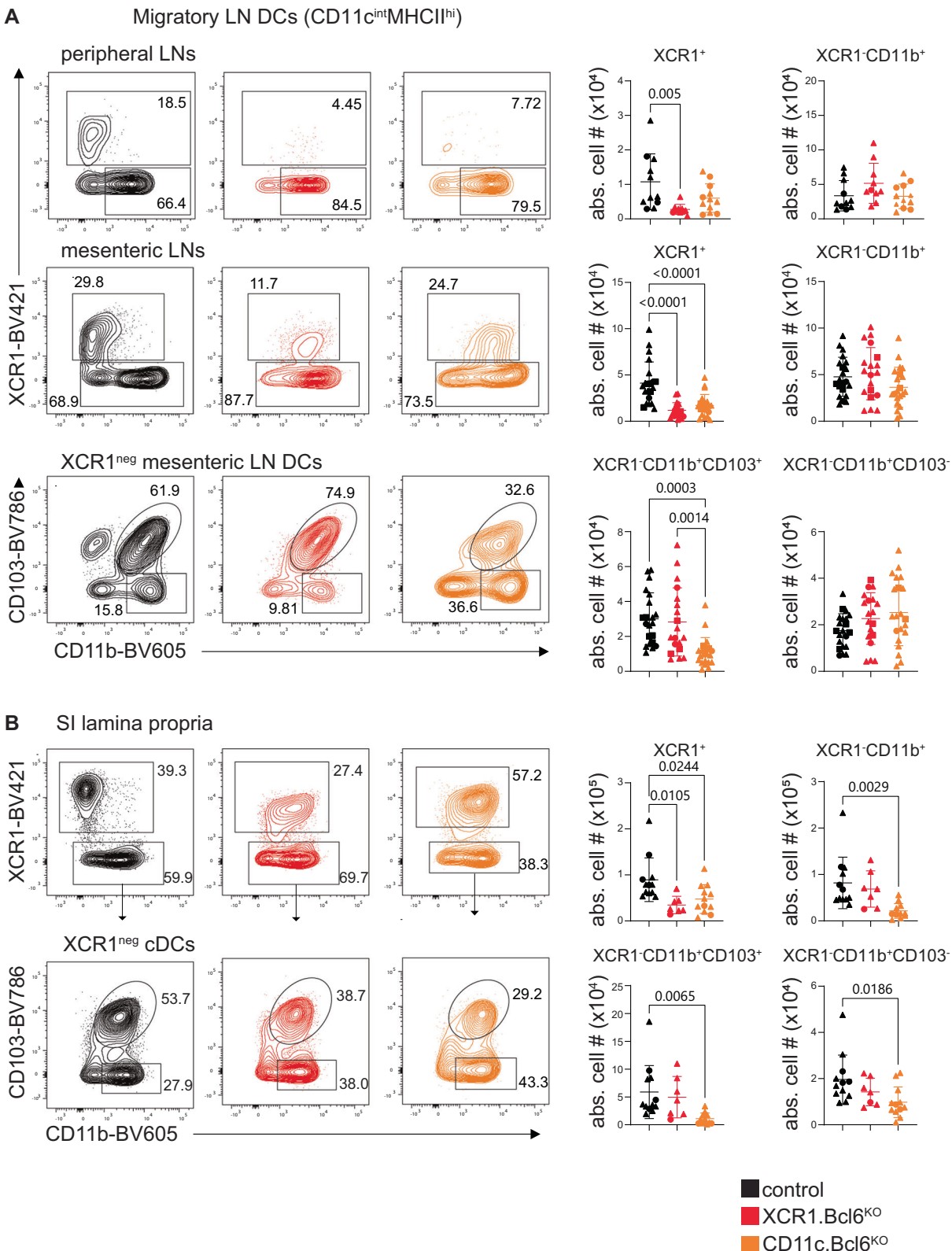

**A** Migratory LN DCs (CD11c$^{int}$MHCII$^{hi}$)

**B** SI lamina propria

control
XCR1.Bcl6$^{KO}$
CD11c.Bcl6$^{KO}$

significantly altered in the absence of Bcl6 (Fig. 3A, C). In total, 607 genes were differentially regulated in XCR1$^+$ cDC from *XCR1.Bcl6$^{KO}$* mice compared with control cDC1, with 473 being upregulated and 134 downregulated. In contrast, of the 452 differentially regulated genes in cDC2 from *CD11c.Bcl6$^{KO}$* mice, only 206 were upregulated, compared with 246 downregulated genes (Fig. 3D, for a list of contributing genes, see Suppl. Data 2). Interestingly, only 79 genes were coordinately

overexpressed in Bcl6-deficient cDC1 and cDC2, and 32 genes were coordinately downregulated in the absence of Bcl6 in both subsets. 12 genes were reciprocally regulated in the two subsets, of which 11 genes were upregulated in cDC1 and downregulated in cDC2 (including receptor-encoding genes *Ffar1* and *Lair1*, enzyme-encoding genes *Tgm2*, *Pigv*, and *Malt1* and genes encoding proteins implicated in cellular and intracellular adhesion and transport *Cdh17*, *Ctnnd2*, *Ston2*,

**Fig. 2 | Phenotype and abundance of migratory cDC subsets lacking Bcl6. A** Left: Flow cytometry plots showing migratory XCR1+ and XCR1−CD11b+ dendritic cell (DC) subsets in the peripheral lymph nodes (LN) (pLN, top) and mediastinal LN (mLN, middle) of control (black), *XCR1.Bcl6^KO* (red), and *CD11c.Bcl6^KO* (orange) mice. mLN XCR1− DC are further sub-divided into CD103+ and CD103− (bottom). Plots show representative staining profiles of live single, CD45+, lineage− (CD3, CD19, CD64, B220, NK1.1), CD11c^int MHCII^hi migratory DC. Right: Absolute numbers of migratory cDC subsets from pLN and mLN of control, *XCR1.Bcl6^KO*, and *CD11c.Bcl6^KO* mice. Statistical analysis by one-way ANOVA, exact *P* values are annotated. Data points represent values from individual mice pooled from 4 (pLN) and 8 (mLN)

experiments with 2–3 (pLN) and 1–6 mice (mLN) per group, and lines represent means ± SD. **B** Left: Flow cytometry plots showing SILP XCR1+ and XCR1−CD11b+ DC subsets from control, *XCR1.Bcl6^KO*, and *CD11c.Bcl6^KO* mice. Plots show representative staining profiles of live, CD45+, lineage− (CD3, CD19, CD64, B220, NK1.1), MHCII+, CD11c^hi gated single cells. Right: Absolute numbers of XCR1+ and XCR1−CD11b+ DC subsets from control, *XCR1.Bcl6^KO*, and *CD11c.Bcl6^KO* mice. Statistical analysis was performed by one-way ANOVA, exact *P* values are annotated. Data points represent values from individual mice pooled from 2 to 3 experiments with 2–3 mice per group, and lines represent means ± SD. Circles: females; triangles: males; squares: sex-information missing.

and *Rin2*). The gene for the actin-binding protein Plectin was the only gene upregulated in cDC2 and downregulated in cDC1. Taken together, although the lineage identity of splenic cDC1 and cDC2 is generally maintained in the absence of Bcl6, this does result in significant changes in subset-specific gene expression in both cDC subsets.

### Bcl6-deficient cDC1 maintain the ability to cross-present antigen

To better characterize the impact of Bcl6-deficiency on cDC1, we next compared the expression of cDC1-related markers using flow cytometry. This confirmed the maintenance of Tlr3, Xcr1, Irf8, CD103, and Clec9a expression by XCR1+ cDC from the spleen of *XCR1.Bcl6^KO* mice, as well as upregulation of CD11b and Sirpα and loss of CD8α (Fig. 4A). In addition, we detected higher expression of the costimulatory ligand CD24 on Bcl6-deficient cDC1.

The most characteristic property of cDC1 is to cross-present exogenous antigen to class I MHC-restricted CD8+ T cells, a function that involves cell−surface receptors such as Clec9A (Dngr1) that mediate the uptake of dead cells[18,19]. Our gene expression analysis suggested that XCR1+ cDC1 maintained the expression of *Clec9a* in the absence of Bcl6, and this was confirmed by flow cytometry (Fig. 4A). We therefore examined whether cross-presentation was intact in *XCR1.Bcl6^KO* animals by injecting heat-killed ovalbumin-expressing mouse embryonic fibroblasts (OVA-MEFs) into recipient mice adoptively transferred with OVA-specific TCR-transgenic CD8+ T cells from OT-I mice. OT-I cell proliferation strictly depends on cDC1-mediated cross-presentation in this model[17]. OT-I cells proliferated to a similar extent in the spleens of OVA-MEF-injected *XCR1.Bcl6^KO* and control mice (Fig. 4B, C), suggesting that Bcl6-deficient cDC1 retain their ability to cross-present. OT-I cell proliferation was also observed in the mLN. In the mLN of *XCR1.Bcl6^KO* mice OT-I cells showed somewhat diminished proliferation compared to control mice. This most likely reflects the decreased number of migratory cDC1 in the mLN of *XCR1.Bcl6^KO* mice (Fig. 2A), as mLN cDC1 from *XCR1.Bcl6^KO* cross-presented antigen with comparable efficiency as control cDC1 in an in vitro cross-presentation assay upon cDC1-input normalization (Suppl. Fig. 4).

### Bcl6-deficiency in cDC1 causes alterations in immunologically relevant pathways

For a broader assessment of whether the absence of Bcl6 influences the immunological functions of cDC1, we next performed gene set enrichment analysis of the RNA-Seq data from XCR1+ cDC in *XCR1.Bcl6^KO* and control mice. This showed that several immunologically relevant gene ontology pathways were enriched within the differentially expressed gene sets, including *leukocyte adhesion*, *proliferation* and *activation*, as well as *production of IL-1* (Fig. 5A). In parallel, KEGG pathway analysis highlighted differences in *cytokine-cytokine receptor interactions*, *NOD-like receptor signaling pathway*, *MERK signaling pathway*, *hematopoiesis cell lineage*, *Tuberculosis* and *C-type lectin receptor signaling pathway* (Fig. 5B, C). Upregulated genes were generally enriched for pro-inflammatory signals, and among others, we found an over-representation of genes involved in the IL-6 pathway in cDC1 from *XCR1.Bcl6^KO* mice (Suppl. Fig. 5). Interestingly, we detected higher levels of serum IL-6 in *XCR1.Bcl6^KO* mice injected with the double-stranded RNA-mimic poly(I:C) when compared to

treated controls (Fig. 5D). These data suggest that Bcl6-mediated transcriptional repression in cDC1 may tune the function of this subset[18,19,17].

Taken together, although cDC1-specific Bcl6 deficiency alters the phenotype and transcriptional profile of cDC1, the hallmark function of cDC1 to cross-present antigen is largely unaffected by the loss of Bcl6.

### Bcl6-deficiency results in a selective loss of Notch2-dependent cDC2

In addition to its role in cDC1, our initial results also revealed a significant role for Bcl6 in the gene expression signature of cDC2. Furthermore, the numbers of cDC2 were reduced in the spleen of *CD11c.Bcl6^KO* mice, as were the numbers of cDC2 expressing CD103 in the intestinal LP and mLNs. Diminished cDC2 numbers in the blood of CD11c.Bcl6^KO mice suggest that this was not due to a release of cDC2 into the bloodstream (Suppl. Fig. 2B). The remaining cDC in the spleen of *CD11c.Bcl6^KO* mice also showed somewhat lower expression of MHCII and higher expression of CD11b than control cDC2 (Fig. 6A). Further phenotyping of spleen cDC2 in *CD11c.Bcl6^KO* mice revealed a striking loss of the ESAM^hi cDC2 subset (Fig. 6B and Suppl. Fig. 2). We verified our findings in an independent mouse model in which Bcl6 deletion was induced by cre recombinase under the control of the *Clec9a* promoter, which targets cDC precursors and their progeny and some pDCs but no other leukocytes[36]. These mice additionally carry a YFP reporter following a floxed stop cassette in the ubiquitous ROSA26 locus to monitor targeting efficiency. As in *CD11c.Bcl6^KO* mice, spleen cDC in *Clec9a.Bcl6^KO* mice showed altered CD11b expression and loss of ESAM^hi cDC2 (Fig. 6C, D and Suppl. Fig. 6). This was apparent in the analysis of bulk cDC2 in *Clec9a.Bcl6^KO* mice and was even more pronounced when pre-gating on YFP+ instead of CD11c, which enriches cDC2 with a history of *Clec9a*-promoter activity (Fig. 6E). Importantly, this also shows that CD11c downregulation in the absence of Bcl6 does not account for the reported phenotype (for a side-by-side comparison of CD11c- and Clec9A-reporter-based gating strategies see Suppl. Fig. 7).

As expected[28], the ESAM^lo subset of cDC2 expressed slightly lower levels of CD11c compared to their ESAM^hi counterparts, and a significant proportion lacked expression of 33D1 and CD4, which were expressed uniformly by ESAM^hi cDC2 (Fig. 6F). The total population of cDC2 remaining in *CD11c.Bcl6^KO* mice resembled the ESAM^lo population in terms of these markers but expressed higher levels of CD11b and IRF4 than either ESAM^hi or ESAM^lo control cDC2 subset, suggesting these markers may be specific targets of Bcl6 repression.

To explore the nature of the cDC2 remaining in the spleen of *CD11c.Bcl6^KO* mice further, we performed gene set enrichment analysis (GSEA) of differentially expressed genes between *CD11c.Bcl6^KO* and control cDC2 in gene sets derived from publicly available bulk RNA-Seq data from ESAM^hi and ESAM^lo cDC2 from control spleen[37]. As expected from their usual predominance in the control spleen, control cDC2 gene signatures were highly enriched in ESAM^hi vs ESAM^lo gene sets, while *CD11c.Bcl6^KO* cDC2 gene signatures were highly enriched in ESAM^lo vs ESAM^hi gene sets (Fig. 7A). We next directly compared the set of genes that were differentially expressed in ESAM^hi vs ESAM^lo cDC2 with the gene set differentially expressed

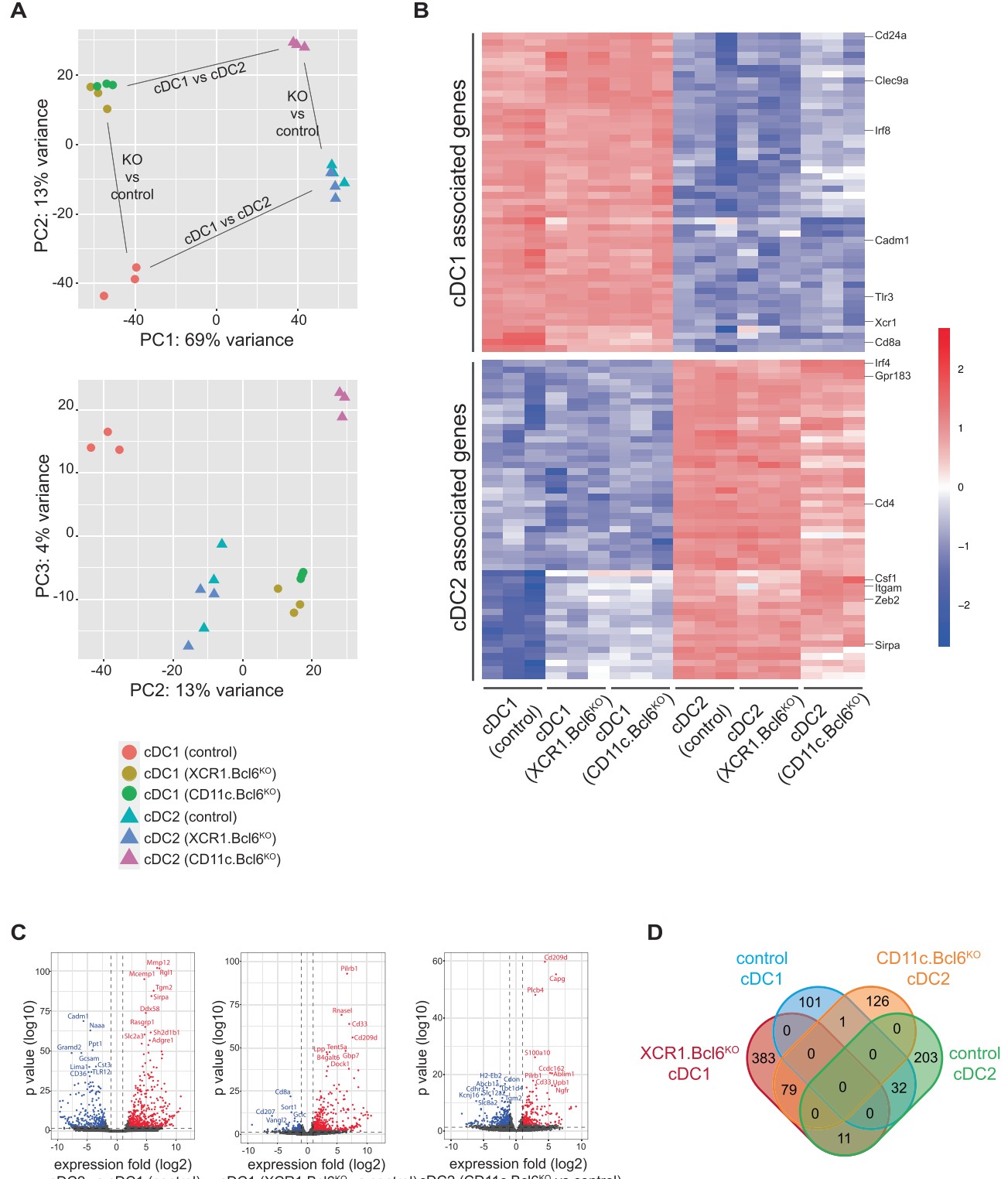

in *CD11c.Bcl6^KO* vs control cDC2. In accordance with the GSEA data, the majority of the top 50 genes expressed more prominently by ESAM^hi cDC2 were also significantly over-represented in the population of total control cDC2, which expressed much fewer of the top 50 genes from ESAM^lo cDC2 (Fig. 7B). Conversely, the genes overexpressed by total cDC2 from *CD11c.Bcl6^KO* spleen showed much more overlap with the top 50 genes expressed by ESAM^lo cDC2, while the genes over-expressed by ESAM^hi cDC2 were mostly downregulated in Bcl6-

deficient cDC2 (Fig. 7B, for a complete gene list containing expression levels, see Suppl. Data 3). These data confirm that Bcl6 deficiency in DC causes the loss of the ESAM^hi lineage of splenic cDC2.

### DC-specific loss of Bcl6 causes defects in Tfh and Tfr cell generation in the spleen

ESAM^hi spleen DC reside in the marginal zone bridging channels, where they play a prominent role in the induction of germinal centers (GC)[29].

**Fig. 3 | Lineage identity of splenic Bcl6-deficient cDC subsets. A** Principal component analysis (PCA) plot of gene expression data obtained from bulk RNA-seq using DESeq2 (PRJNA834905). Color-coded icons show classical dendritic cells (cDC)1 and cDC2 from the spleen of control, *XCR1.Bcl6$^{KO}$*, and *XCR1.Bcl6$^{KO}$* mice. Three replicates were included in each group. **B** Heatmap of selected highly expressed cDC1 or cDC2 associated genes in all RNA-Seq samples (50 most differentially expressed genes between cDC1 and cDC2 chosen from the 100 genes with highest expression level among total differentially expressed genes (fold change ≥ 2, padj: 0.01, calculated by DESeq2 using Benjamini−Hochberg corrections of two-sided Wald test *P* values) in control samples). The color scale represents the row *Z* score. **C** Volcano plots showing the global transcriptional differences between splenic cDC2 and cDC1 in control (left), *XCR1.Bcl6$^{KO}$* (middle) and *CD11c.Bcl6$^{KO}$* mice (right). Each circle represents one gene. The log2FoldChange (LFC) is represented on the *x*-axis. The *y*-axis shows the −(log10 of the *p* adjust value). A padj value of 0.01 (calculated by DESeq2 using Benjamini−Hochberg corrections of two-sided Wald test *P* values) and an absolute LFC of 1 is indicated by dashed lines. The upregulated or downregulated genes are shown in red or blue dots, respectively, and the top-ranked gene symbols with the lowest padj value were annotated in the plot. **D** VENN diagram showing significantly elevated differentially expressed genes (DEG) (|Log2foldchange | ≥1 and padj < 0.01) between *XCR1.Bcl6$^{KO}$* cDC1 (red) and control cDC1 (blue), and between *CD11c.Bcl6$^{KO}$* cDC2 (orange) and control cDC2 (green).

Confocal microscopy using CD11c as a pan-DC marker and TLR3 and Sirpα to define cDC1 and cDC2, respectively, confirmed a substantial loss of DC from this region in the *CD11c.Bcl6$^{KO}$* spleen (Fig. 8A). This was also evident in *Clec9a.Bcl6$^{KO}$* and *Clec9a.Bcl6$^{WT}$* control mice carrying a YFP reporter under the control of the *Clec9a* promoter, in which Bcl6-deficiency in DCs led to a similar decrease in marginal zone bridging channel DCs (Fig. 8B). To test the functional relevance of this defect, we adoptively transferred OVA-specific TCR-transgenic CD4$^+$ T cells (OT-II) into control and *CD11c.Bcl6$^{KO}$* mice immunized the mice with OVA + poly(I:C) intraperitoneally and assessed the generation of antigen-specific Tfh cells three days later (Suppl. Fig. 8A). We chose this timepoint because this early Tfh cell induction is independent of GC B cells[38,39], which are known to express CD11c and are required for the maintenance of Tfh cells at later times[40]. Although OT-II cells expanded normally in the *CD11c.Bcl6$^{KO}$* mice, in response to immunization, the early induction of antigen-specific CXCR5$^+$Bcl6$^+$ Tfh cells was markedly reduced in immunized *CD11c.Bcl6$^{KO}$* mice compared to control mice (Suppl. Fig. 8A), suggesting that Bcl6-dependent cDC indeed drive early Tfh cell induction in this model. To enable analysis of later time points of the Tfh cell response and to confirm our findings in mice in which Bcl6 deletion is confined to cDC, we next adoptively transferred naïve OT-II cells into *Clec9a.Bcl6$^{WT}$* control or *Clec9a.Bcl6$^{KO}$* mice, followed by i.p. immunization with NP-OVA in alum adjuvants and analysis of the spleen seven days later (Fig. 8C). The frequency of CXCR5$^+$PD-1$^+$ and CXCR5$^+$Bcl6$^+$ Tfh cells was significantly reduced in OT-II cells recovered from *Clec9a.Bcl6$^{KO}$* hosts as compared to control animals (Fig. 8D and Suppl. Fig. 8B). To better assess the long-term effects of DC-restricted Bcl6 deficiency on Tfh cell and humoral immune responses, we immunized *Clec9a.Bcl6$^{WT}$* and *Clec9a.Bcl6$^{KO}$* mice i.p. with NP-KLH/alum and analyzed the spleens two weeks later (Fig. 8E). Total spleen cellularity was not significantly increased in *Clec9a.Bcl6$^{KO}$* mice (Fig. 8F). As regulatory T (Treg) cells do not develop from naive OT-II precursor cells, such as in the previous adoptive transfer experimental setting, and Tfr cells share similarities with Treg cells and Tfh cells[41,42], such as expression of Foxp3 and Bcl6, respectively, we quantified all three subsets within the endogenous T cell compartment following NP-KLH/alum immunization. Total numbers of Foxp3$^−$ and Foxp3$^+$ CD4$^+$ T cells were not affected by DC-restricted Bcl6-deficiency (Fig. 8F and Suppl. Fig. 8C). While the frequencies of endogenous CXCR5$^{int}$PD-1$^{int}$ and CXCR5$^{int}$Bcl6$^{int}$ Tfh cells were reduced among conventional Foxp3$^−$ CD4$^+$ T cells of the conditional knockout mice, the frequencies of GC Tfh cells, gated as CXCR5$^{hi}$PD-1$^{hi}$ or CXCR5$^{hi}$Bcl6$^{hi}$, were not altered in *Clec9a.Bcl6$^{KO}$* mice at day 14 after immunization (Fig. 8G, H, and Suppl. Fig. 8C). In contrast, we observed a strong reduction in CXCR5$^{hi}$PD-1$^{hi}$ or CXCR5$^{hi}$Bcl6$^{hi}$ Tfr cells among Foxp3$^+$ CD4$^+$ T cells (Fig. 8G, H, and Suppl. Fig. 8C), indicating that Bcl6 in DCs is particularly important for the generation of Tfr cells. To address the impact of DC-restricted Bcl6-deficiency on the antibody response, we assessed the NP-specific serum antibody levels at different time points following NP-KLH/alum immunization (Fig. 8 I), focusing on total IgG and the signature isotype induced by type-2 immunization with alum adjuvants, IgG1. Quantifying both the amounts of NP-specific antibodies against NP$_{36}$-coated BSA, a proxy for total NP-binding antibodies, as well as against NP$_2$-coated BSA, a proxy for high-affinity NP-binding antibodies, we found reduced NP-specific antibody responses in *Clec9a.Bcl6$^{KO}$* mice (Fig. 8J and Suppl. Fig. 8D). In conclusion, Bcl6 deficiency in cDC causes the loss of ESAM$^{hi}$ cDC2 and impairs Tfh, Tfr, and antibody responses in the spleen.

## DC-specific loss of Bcl6 causes defects in Th17 cell priming in the colon of mice infected with *C. rodentium*

In addition to their role in driving GC reactions in the spleen, Notch2-dependent cDC2 have been shown to support Th17-dependent immunity towards *C. rodentium* in the colon[22]. The colonic LP (cLP) of *CD11c.Bcl6$^{KO}$* mice contained significantly fewer cDC than control mice, with the most pronounced loss detected in the CD103$^+$ cDC2 population (Fig. 9A, B). To analyze whether Bcl6-deficiency recapitulated the functional consequences of Notch2-deficiency, we infected *CD11c.Bcl6$^{KO}$* and control littermates with *C. rodentium*. *CD11c.Bcl6$^{KO}$* mice lost comparable weight to control littermates early after infection, with delayed recovery as evidenced by significantly lower body weight from day 11 onwards (Fig. 9C). Consistent with a lack in weight difference between genotypes at day nine post-infection, colon length, and bacterial load were comparable at this time point, while there was a tendency towards shorter colons and elevated bacterial load in *CD11c.Bcl6$^{KO}$* mice compared to control littermates 21 days post-infection (Fig. 9C). At this late stage during infection, colons from *CD11c.Bcl6$^{KO}$* mice continued to show mild pathology, while control littermates appeared fully healed (Fig. 9D). In parallel, there were blunted Th17 responses in the cLP of *CD11c.Bcl6$^{KO}$* mice nine days after *C. rodentium* infection (Fig. 10A, gating strategies depicted in Suppl. Fig. 9), while Th1 and Treg numbers were not affected. This defect in Th17 priming was no longer evident 21 days after infection (Fig. 10B). To address whether our results were due to a lack of Bcl6 in bona fide DCs, we infected *Clec9a.Bcl6$^{KO}$* mice and controls with *C. rodentium* and analyzed their CD4 T cell compartment on day 9 post-infection. While we detected a trend towards lower RORγt$^+$ CD4 T cell numbers, Tbet$^+$ and Foxp3$^+$ CD4 T cells were more prominently affected (Suppl. Fig. 10A). Analysis of YFP expression as a surrogate of *Clec9a.cre* activity in intestinal DCs in *Clec9a.Bcl6$^{KO}$* mice, however, revealed only partial targeting of cDC2 in the gut, potentially explaining the moderate deficiency in Th17 priming (Suppl. Fig. 10B).

These data show that Bcl6-deficiency in CD11c$^+$ mononuclear phagocytes causes the loss of a prominent colonic cDC2 subset, with a consequent defect in Th17 priming in response to *C. rodentium* infection, leading to delayed healing and recovery.

## Discussion

We show here that Bcl6-deficiency in the DC compartment causes major transcriptional alterations with subset-specific consequences. Specifically, Bcl6-deficient cDC1 acquire an XCR1/CD11b double-positive phenotype not seen in control mice, where XCR1 and CD11b are mutually exclusive markers of cDC1 and cDC2, respectively. However, Bcl6-deficient cDC1 retained a cDC1-like transcriptome, cDC1 lineage markers, and the ability to cross-present. The lack of Bcl6 in DC also revealed a critical role for Bcl6 in the development of the ESAM$^{hi}$

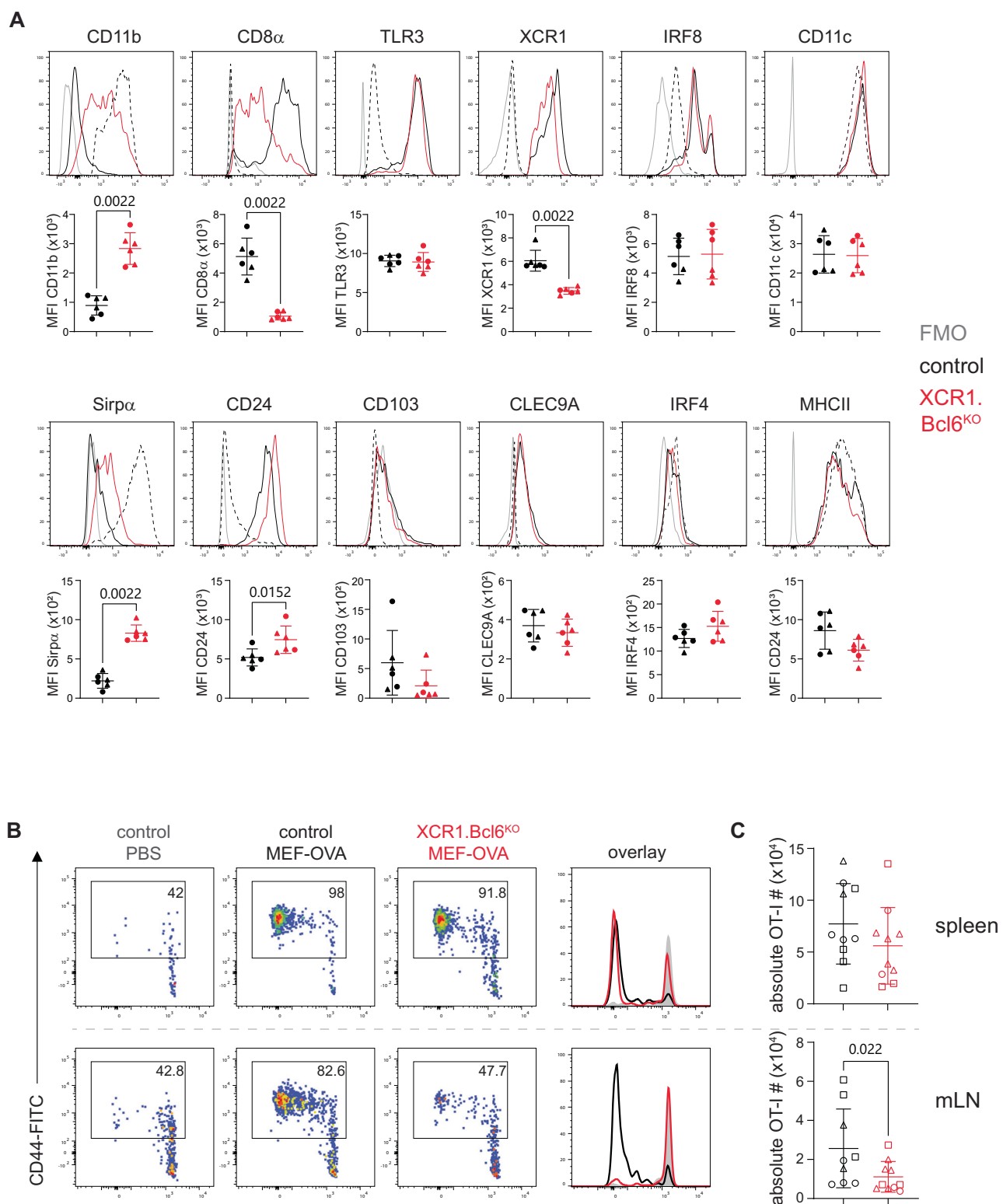

subset of cDC2 in the spleen and CD103⁺CD11b⁺ cDC2 in the intestines, and there was a parallel defect in splenic Tfh cell induction in response to protein antigen immunization and in intestinal Th17 induction in response to *C. rodentium* infection. Together these findings extend the role of Bcl6 to the regulation of cDC2 behavior in GC function and the periphery.

We previously described a selective loss of putative cDC1 in the spleen of Bcl6-deficient mice based on the absence of CD8α⁺CD11b⁻ cDC[20]. Here we refine this interpretation by showing that the numbers of resident cDC1 are, in fact normal when Bcl6 is lacking in CD11c⁺ cells or cDC1, but they upregulate CD11b and down-regulate CD8α, causing them to merge into the classical cDC2 gate. In contrast, the numbers of

**Fig. 4 | Phenotype and cross-presentation abilities by cDC1 deficient in Bcl6.**
**A** Flow cytometric analysis of classical dendritic cells (cDC)1 in the spleen of control and *XCR1.Bcl6^KO* mice. Representative histograms show indicated protein expression by control (black) and *XCR1.Bcl6^KO* (red) cDC1, with fluorescent minus one (FMO) staining shown in gray, and control cDC2 is shown as a dashed line. Dot plots show FMO-subtracted mean fluorescent intensity (MFI) values. Pre-gating on cDC1 and cDC2 was performed as in 1A. Histograms are representative of 3 independent experiments with 3 mice each MFI plots show 2 out of 3 experiments, and lines represent means ± SD. Statistical analysis using the Mann–Whitney test, exact *P* values are annotated. **B** Representative flow cytometry plots showing cell tracer violet (CTV) dilution and CD44 expression on OT-I cells (gated on live,

CD3+CD8+CD45.1+ single cells) in the spleen (top) and mesenteric lymph nodes (mLN, bottom) of control and *XCR1.Bcl6^KO* recipient mice 3 days after i.p. immunization with heat-shocked ovalbumin-expressing mouse embryonic fibroblasts (OVA-MEF). Data are representative of 3 independent experiments with 2–5 mice each. **C** Total number of OT-I cells in the spleen (right) and mLN (left) of immunized control and *XCR1.Bcl6^KO* mice. Data are pooled from 3 independent experiments with 1–4 mice each. Data points represent values from individual mice mouse, and lines represent means ± SD. Statistical analysis using the Mann–Whitney test, exact *P* values are annotated. Circles: females; triangles: males; squares: sex-information missing.

---

migratory CD11b− CD103+ cDC1 are decreased in the mLN and intestinal mucosa of mice lacking Bcl6 within the DC compartment, a finding supported by defective cross-presentation activity in the mLNs. We currently cannot explain the differential effects of Bcl6 deficiency on resident versus migratory cDC1 populations. However, a similar dichotomy has been described recently for dependence on DC-SCRIPT, whose absence predominantly diminished resident cDC1[43]. As Bcl6 primarily acts as a transcriptional repressor, it is possible that DC-SCRIPT and Bcl6 integrate into a feedback loop to regulate cDC1 migration or survival upon maturation, and this idea requires further study.

As Bcl6 is not expressed by DC progenitors in the bone marrow and their frequency is not affected by the lack of Bcl6[35,44], Bcl6 seems to act downstream of other cDC1 lineage specifying TFs such as IRF8, Nfil3, BATF3, DC-SCRIPT, and Id2. In models deficient of those TFs, the cDC1 lineage is affected from the precursor level onwards, preventing their expression of all lineage markers, as well as their ability to cross-present antigen[10,12–14,43,45–48], hallmarks that were preserved in Bcl6-deficient cDC1. Rather, it appears that Bcl6 may fine-tune cDC1 identity by suppressing selected cDC2 genes at the mature state, and as a result, Bcl6-deficient cDC1 maintain key cDC1 genes such as XCR1, TLR3, and CLEC9A as well as cross-presentation function. However, they show a gene expression profile that suggests a cDC2-related "gain-of-function", with enhanced expression of genes of the cytokine-cytokine receptor pathway, cell adhesion and migration pathways, and pathways associated with inflammasome activity. Interestingly, cDC1 with upregulated CD11b and Sirpα have also been described when the TF ZEB2 is over-expressed from the pre-cDC stage onwards[25]. Exploring how Bcl6 interacts with this and other lineage-determining factors to regulate cDC1 gene expression and functions will be an important topic for future research.

*CD11c.Bcl6^KO* and *Clec9a.Bcl6^KO* mice lacked the dominant subset of ESAM^hi cDC2 in the spleen, and a comparison of gene expression profiles between ESAM^hi and ESAM^lo control cDC2 subsets and cDC2 from *CD11c.Bcl6^KO* mice showed that Bcl6-deficient cDC2 closely aligned with control ESAM^lo cDC2. However, this overlap was not absolute, suggesting that Bcl6-deficiency causes transcriptional changes in all cDC2. For example, the extent of dysregulation of IRF4 and CD11b expression in the absence of Bcl6 cannot solely be explained by the loss of ESAM^hi cDC2. In addition, both markers were also upregulated in Bcl6-deficient cDC1, again indicating a wider role for Bcl6 in control of mature cDC differentiation. Interestingly, the consequences of Bcl6-deficiency on cDC overlap with those previously shown for Notch2-deficiency[20,28,34]. Specifically, both strains show deficiencies in cDC1 and ESAM^hi cDC2, but it remains to be determined whether the apparent loss of cDC1 in Notch2-deficient animals reflects the genuine absence of these cells, or is because their phenotype has simply changed, as we show in Bcl6-deficient mice. Of note, the cDC1 marker CD24 is upregulated in both Bcl6- and Notch2-targeted mice (our study and[22,28]), arguing against a developmental defect within the cDC1 spleen compartment in Notch2-targeting models. It will be of interest to explore how the Notch2- and Bcl6-pathways interact in DC biology.

Consistent with the findings that cDC2 equivalent to the ESAM^hi population located preferentially in marginal zone bridging channels in the spleen[49], the dysregulation in splenic cDC populations in both Bcl6- and Notch2-deficient mice was associated with a loss of cDC from this region of the spleen ([29] and current results). Localization of cDC2 to the bridging channels is dependent on the homing receptor GPR183[50,51] and Bcl6 regulates GPR183 expression in B cells[52]. However, GPR183 expression levels were not altered in Bcl6-deficient cDC2, suggesting either that additional factors are required for the correct positioning of cDC2 in the spleen or that Bcl6-deficiency causes a development block or diminished survival within ESAM^hi cDC2 irrespective of their location. Along those lines, GPR183-deficiency in cDC2 specifically diminished antibody response generation to particulate antigen, while responses to soluble antigen were unaffected[51]. The defect in ESAM^hi cDC2 we found in *CD11c.Bcl6^KO* and *Clec9a.Bcl6^KO* mice were accompanied by reduced induction of Tfh cells in response to parenteral immunization with soluble OVA, suggesting a general deficiency in supporting GC reactions in these mice. While it has been suggested that Notch2 is needed to ensure correct positioning of cDC2 for access to antigen in the bloodstream[28], it has to our knowledge, not been addressed whether response deficits were limited to particulate antigen in these mice. Further investigation is needed to explore the importance of correct localization on development, survival, and antigen capture by cDC2 and how this might be controlled by a Bcl6–Notch2 axis.

DC-specific deletion of Bcl6 caused alterations in the generation of Tfh cells, Tfr cells, and corresponding antibody responses. As T and B cell-intrinsic expression of Bcl6 is known to be required for Tfh cell and GC B cell differentiation[33,53], our findings extend this concept to the DC stage. Our data indicate that Tfh and Tfr cell fates of activated conventional CD4+ T cells and Treg cells, respectively, are pre-determined or imprinted by Bcl6 expression patterns already in DCs. One interesting finding was the strong effect of DC-specific Bcl6 deficiency on Tfr cells, which are believed to mainly develop from thymic Tregs[41,42]. Given the stronger reduction in Tfh cells in *Clec9a.Bcl6^KO* mice at early stages following immunization (days three and seven), as compared to later stages (day 14), it is possible that a continued lack of Tfr-mediated suppression of Tfh cells may allow for a compensatory outgrowth of Tfh cells over time, thus mitigating the net effects of DC-specific Bcl6 deficiency. This would be in line with the multistep differentiation process of Tfh cells in which B cells become the sole antigen-presenting cell subset in GCs, with no classical DCs being present in GCs[54].

XCR1− CD11b+CD103+ intestinal LP DC were also reduced in the absence of Bcl6 in DC, again mirroring the phenotype observed in Notch2-deficient mice[22,28]. cDC2 are potent drivers of Th17 responses in the intestines[23,55], but recent findings suggest that DC-independent Th17 cell induction can also occur[56]. As *CD11c.Notch2^KO* mice succumb early to *C. rodentium* infection due to deficient IL-23-driven innate immunity, the adaptive immune response has not been investigated in this model[22]. In contrast, the early stages of infection were normal in *CD11c.Bcl6^KO* mice, but these mice showed a delay in recovery at later stages and altered kinetics of Th17 induction, with significantly lower

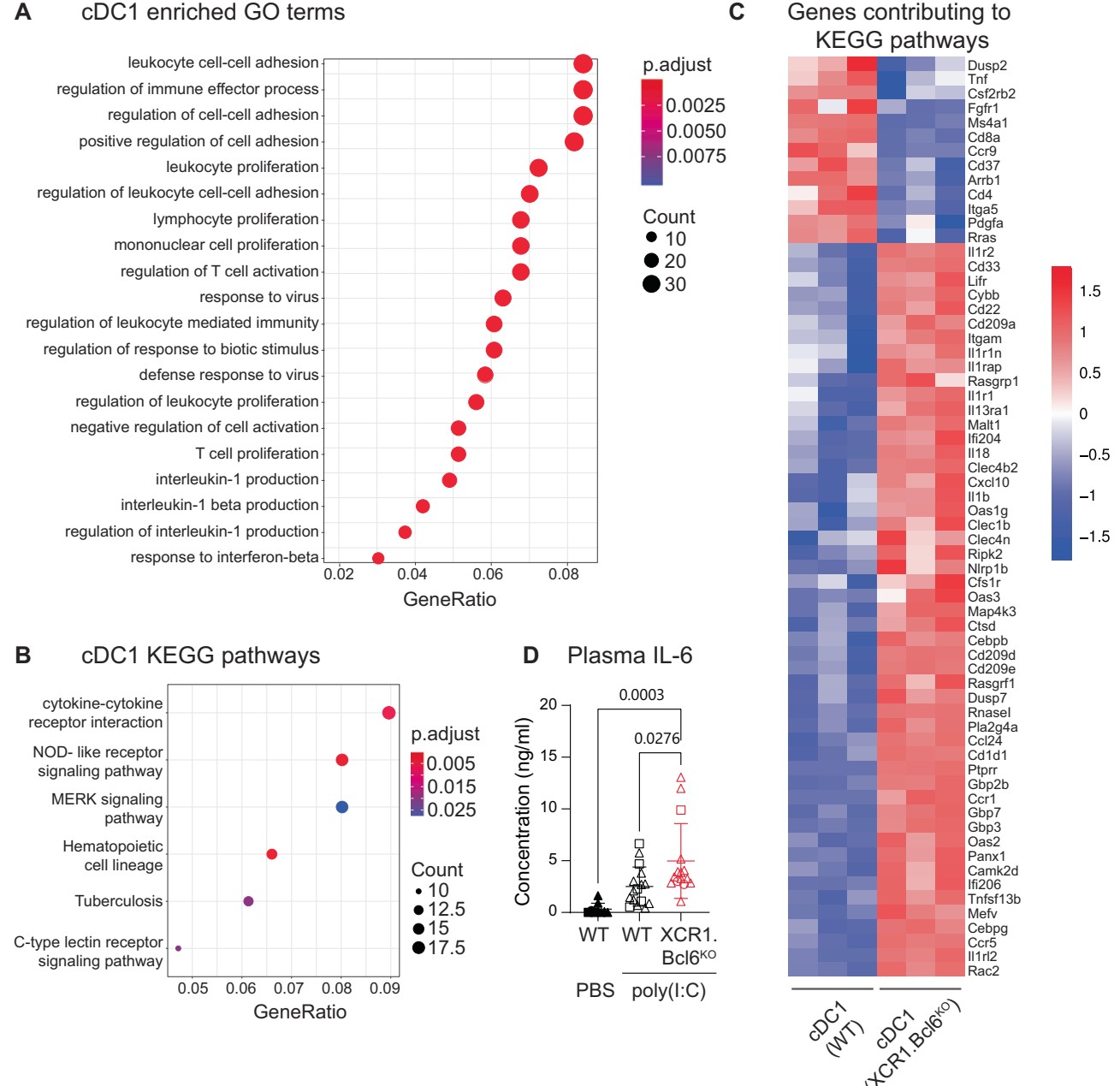

**Fig. 5 | Differentially expressed genes in Bcl6-deficient splenic cDC1. A** Gene ontology (GO) term enrichment analysis of differentially expressed genes (DEG) in *XCR1.Bcl6^KO* and control cDC1 (PRJNA834905). The *p* value calculated from GO term enrichment analysis is represented on the *x*-axis, and the y-axis shows the GO Biological Process (BP) terms. *P*-values were calculated by using enrichGO function from R package clusterProfiler with the one-sided hypergeometric test. **B** KEGG (Kyoto Encyclopedia of Genes and Genomes) term enrichment analysis of DEGs in *XCR1.Bcl6^KO* vs control cDC1. The GeneRatio value calculated from KEGG term enrichment analysis is represented on the *x*-axis, and the *y*-axis shows the KEGG enriched terms. **C** Heatmap of all genes contributing to the enriched KEGG pathways depicted in (**B**). The color scale represents the row *Z* score. **D** IL-6 levels in the serum of *XCR1.Bcl6^KO* vs control mice injected intraperitoneally with poly(I:C) 2 h before analysis. Data are pooled from 6 experiments with 1–6 mice each (2 experiments without untreated controls), and lines represent means ± SD. Statistical analysis was performed by one-way ANOVA (and Tukey's multiple comparison test), and exact *P* values were annotated. Circles: females; triangles: males; squares: sex-information missing.

Th17 numbers at day nine, which had normalized by day 21. We cannot currently explain the differences in the early innate immune response between *CD11c.Notch2^KO* and *CD11c.Bcl6^KO* mice, but our data are consistent with the idea that DCs play the dominant role in coordinating the adaptive immunity required to terminate the infection. However, while Bcl6-dependent cDC2 support optimal Th17 induction to *C. rodentium*, they are not essential for this response, as Th17 numbers normalized over time, possibly due to continuous proliferation in the presence of the pathogen.

Our results show that Bcl6 deficiency does not alter cDC1 and cDC2 lineage identity, but it alters their transcriptional profile and relative abundance of both lineages and their subsets in different tissues. Specifically, although cDC1 in all tissues examined acquired several cDC2 markers in the absence of Bcl6, they retained their transcriptional and functional identity as cDC1. In contrast, Bcl6 deficiency had differing effects on cDC2 subsets that depended on their location and phenotype, with selective depletion of ESAM^hi cDC2 in spleen and CD103⁺CD11b⁺ cDC2 in the intestine. These effects were

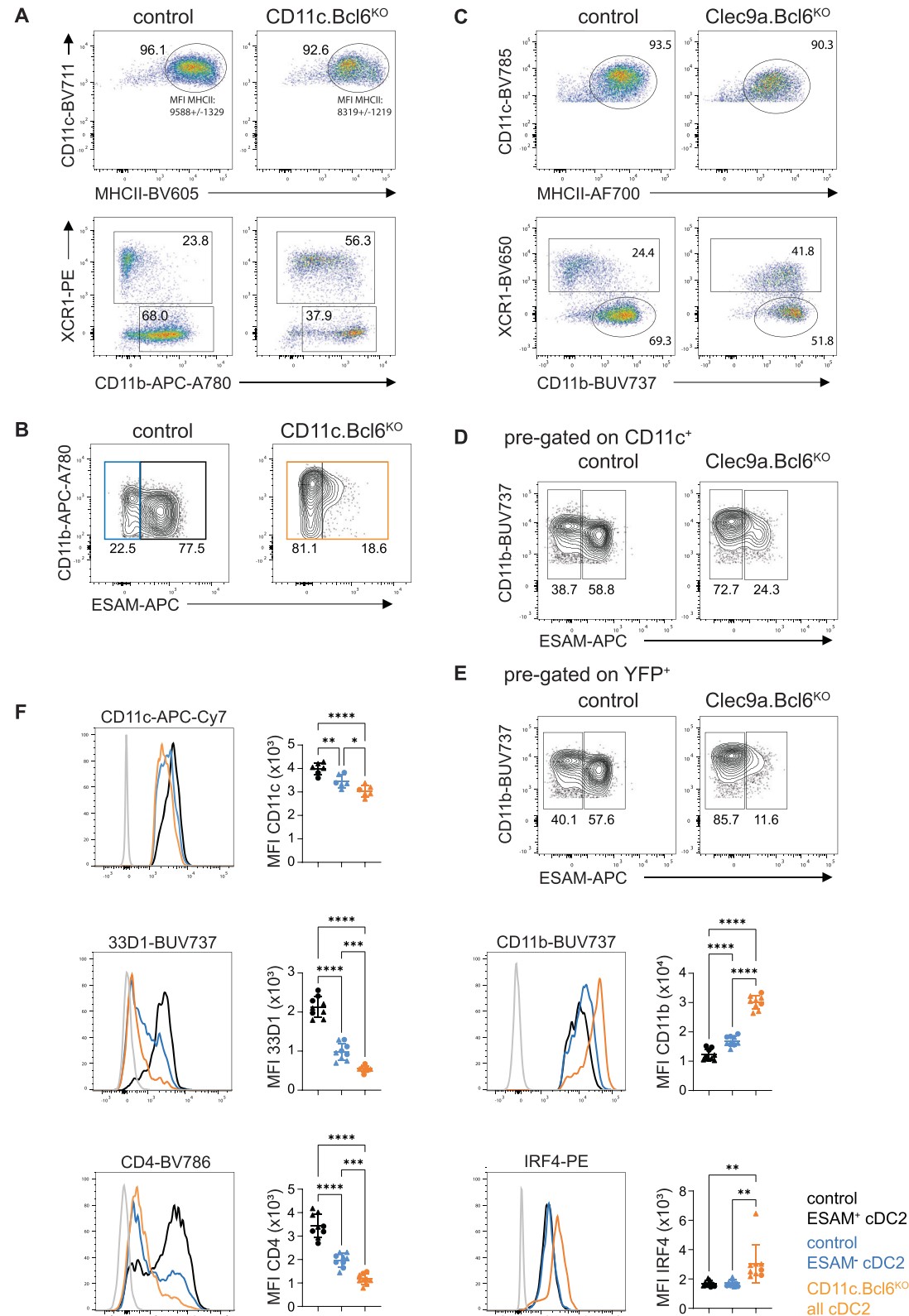

similar to those described for a deficiency in Notch2 and had significant functional consequences for adaptive immunity in the spleen and intestine. These data add to the emerging and poorly understood heterogeneity within the cDC2 compartment by establishing that certain cDC2 subsets require Bcl6 expression in the DC compartment in addition to Notch2- and lymphotoxin β-signaling. Finally, complementary to its established roles in controlling B and T cell-intrinsic cell fate decisions[53], our findings establish Bcl6 as a subset-specific regulator of DC development and function with critical consequences for the induction of adaptive immune responses.

## Methods

All reagents, including manufacturer, catalog number, and dilutions, are listed in Suppl. Data 4.

**Fig. 6 | Phenotype and gene expression profiles of splenic Bcl6-deficient cDC2.**
**A** Representative flow cytometric analysis of classical dendritic cells (cDC)2 in the spleen of control and *CD11c.Bcl6^{KO}* mice, showing expression of CD11c vs MHCII (top panels) and XCR1 vs CD11b (lower panels) by live, lineage⁻ (CD3, CD19, CD64, B220, NK1.1), CD11c⁺ single cell-gated splenocytes. Data are representative of at least 8 independent experiments with 1–3 mice each. Reported mean fluorescent intensity (MFI) levels of MHCII are raw values and average deviation from 3 experiments with 3 mice each. **B** Representative flow cytometric analysis of ESAM^{hi} and ESAM^{lo} cDC2 in the spleen of control and *CD11c.Bcl6^{KO}* mice. Cells were pre-gated as XCR1⁻CD11b⁺ cDC2 from live, lineage⁻ (CD3, CD19, CD64, B220, NK1.1), CD11c⁺, MHCII⁺ single cells. Data are representative of 3 independent experiments with 1–3 mice each. **C** Representative flow cytometric analysis of cDC2 in the spleen of control and *Clec9a.Bcl6^{KO}* mice, showing expression of CD11c vs MHCII (top panels) and XCR1 vs CD11b (lower panels, gated on CD11c⁺, MHCII⁺) by live, lineage⁻ (CD3, CD19, CD64, B220, NK1.1) single cell-gated splenocytes. Data are

representative of 2 independent experiments with 3 mice each. **D** Representative flow cytometric analysis of ESAM^{hi} and ESAM^{lo} cDC2 in the spleen of control and *Clec9a.Bcl6^{KO}* mice. Cells were pre-gated as XCR1⁻CD11b⁺ cDC2 from live, lineage⁻ (CD3, CD19, CD64, B220, NK1.1), CD11c⁺, MHCII⁺ single cells (top). Data are representative of 2 independent experiments with 3 mice each, and lines represent means ± SD. **E** Same data as in **D**, but pre-gated on YFP⁺ instead of CD11c⁺ (see Suppl. Fig. 4 for full gating strategy). Data are representative of 2 independent experiments with 3 mice each, and lines represent means ± SD. **F** Representative flow cytometric analysis of expression of CD11c, 33D1, CD4, CD11b, and IRF4 by splenic XCR1⁻CD11b⁺ cDC2 from control (ESAM^{hi}: black, ESAM^{lo}: blue) or *CD11c.Bcl6^{KO}* mice (orange). FMO controls are depicted in gray. Gates correspond to those depicted in Fig. 6B. Results are representative of at least 2 (CD11c) or 3 (all others) independent experiments with 3 mice each, and lines represent means ± SD. Statistical analysis was performed by one-way ANOVA. *$P < 0.05$, **$P < 0.01$, ***$P < 0.001$ and ****$P < 0.0001$. Circles: females; triangles: males.

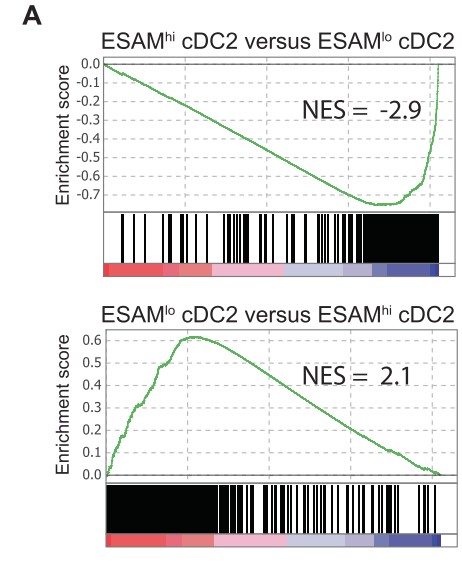

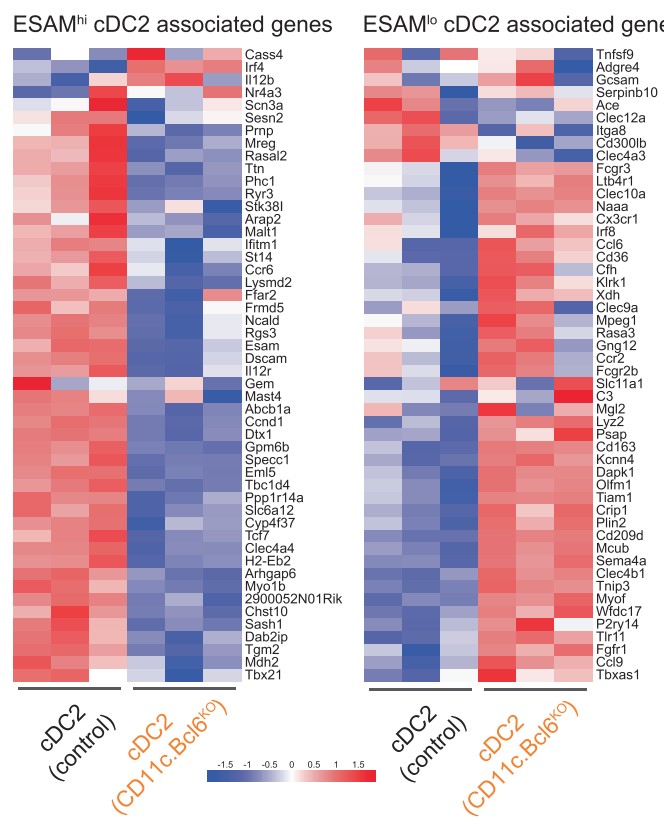

**Fig. 7 | Gene signature comparisons between ESAM^{hi} and ESAM^{lo} cDC2 with Bcl6-deficient cDC2. A** Gene set enrichment plots following *CD11c.Bcl6^{KO}* vs control classical dendritic cells (cDC)2 in an ESAM^{hi} cDC2 related gene set (top) vs an ESAM^{lo} cDC2 related gene set (bottom) (ESAM^{hi} and ESAM^{lo} gene set derived from GSE76132). NES normalized enrichment score. False discovery rate (FDR) $q$-value:

0.0. **B** The top 50 genes associated with spleen ESAM^{hi} cDC2 (left) or ESAM^{lo} cDC2 (right) were derived from GSE76132. Heatmaps show the expression of these genes in the RNA-Seq analysis of cDC2 from control and *CD11c.Bcl6^{KO}* spleen. The color scale represents the row $Z$ score (Expression $Z$-scale).

## Mice

*Xcr1.cre* (*B6-Xcr1^{tm2Ciphe}*)[57], *CD11c.cre*[31], and *Clec9a.cre*[36] mice were crossed to *Bcl6^{fl/fl}* mice[58] (from Alexander Dent or JAX.org stock number #023727) to obtain DC-specific Bcl6-knockout models. *Clec9a.Bcl6^{KO}* mice and *Clec9a.Bcl6^{WT}* control mice were both maintained homozygous for cre and additionally contained homozygous a *Rosa26^{fl-Stop-fl-YFP}* allele (JAX.org stock number 006148). *CD11c.Bcl6* and *XCR1.Bcl6* models were maintained heterozygous for cre and negative littermates were used as control controls. Additional strains used in this study were B6.SJL-Ptprc^a Pepc^b/BoyJ (congenic C57BL/6.CD45.1 mice from the

Jackson Laboratory), C57BL/6-Tg (TcraTcrb)1100Mjb/J and B6.Cg-Tg(TcraTcrb)425Cbn/J (OT-I and OT-II mice from Jackson Laboratory). Mice were group-housed in individually ventilated cages, maintained on a 12 h light and dark cycle at 22 °C and 55% humidity, and maintained under specific pathogen-free conditions. Euthanization was performed using cervical dislocation without anesthesia. Experimental groups were sex- and age-matched. Male and female littermates were used between 8 and 15 weeks of age, and RNA-Seq was performed on female mice only. All animal experiments were performed in accordance with European regulations and federal law of Denmark and Germany.

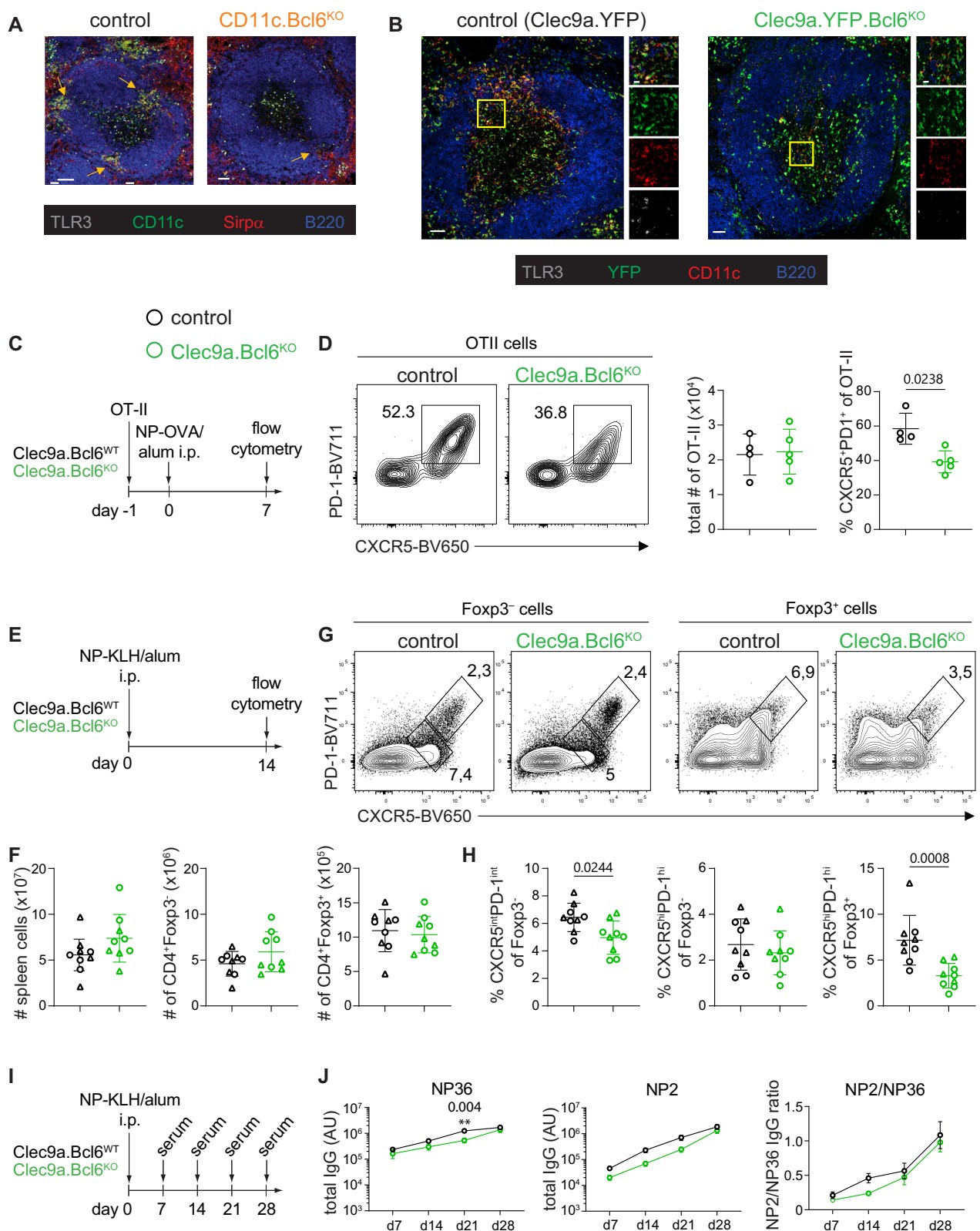

Permission was granted either by the Danish Animal Experiments Inspectorate, the Regierung von Oberbayern, or the Landesamt für Natur-, Umwelt und Verbraucherschutz NRW.

**Cell preparation**

Spleens, mLNs, and pLN (axillary, brachial, and inguinal) were digested in RPMI containing 5% FCS, 0.5 mg/ml Collagenase IV (USBiological), and 30 µg/ml DNase I grade II (Roche) for 45 min at room temperature. Red blood cells in the spleen and blood samples were subsequently lysed using red blood cell lysis buffer (ammonium chloride, potassium bicarbonate, EDTA, and MiliQ water) for 3 min at room temperature. For the OT-II adoptive transfer experiments, splenic single-cell suspensions were prepared and analyzed as previously described[59].

**Fig. 8 | Impact of Bcl6 deficiency in splenic cDC on Tfh cell and antibody responses. A** Confocal microscopy of spleens from *CD11c.Bcl6^KO* and control mice. TLR3 (gray), CD11c (green), Sirpα (red), B220 (blue). Scale bar = 50 μm. Arrows indicate marginal zone bridging channels. Images are representative of 4 (*CD11c.Bcl6^KO*), or 3 (control) replicates from individual mice. **B** Confocal microscopy of spleens from *Clec9a.Bcl6^WT and Clec9a.Bcl6^KO* mice, each carrying homozygous *Rosa26^fl-Stop-fl-YFP* alleles. TLR3 (gray), YFP (green), CD11c (red), B220 (blue). Scale bar = 50 μm. Single channel figures show areas in the yellow boxes. Images are representative of 4 replicates from individual mice. **C** Experimental design. **D** Representative flow cytometry of splenic OT-II (CD45.1/2) cells from *Clec9a.Bcl6^WT* and *Clec9a.Bcl6^KO* recipients 7 days after i.p. immunization with NP-OVA. Tfh cells were gated as CXCR5⁺PD-1⁺ live CD4⁺CD19⁻ lymphocytes. Statistical analysis was performed by two-way Mann–Whitney, exact P values are annotated. Each symbol represents one biological replicate (*n* = 4–5), and lines represent means ± SD. Data are representative of 2 independent experiments. **E** Experimental design. **F** Total cellularity and number of CD4⁺Foxp3⁻ and CD4⁺Foxp3⁺ cells in

*Clec9a.Bcl6^KO* and control mice 14 days after i.p. injection with NP-KLH and alum. Data are pooled from 2 independent experiments with 4-5 mice per group, and lines represent means ± SD. **G** Representative flow cytometry of Foxp3⁻ (left) and Foxp3⁺ (right) CD4⁺ T cell-gated splenocytes 14 days post-NP-KLH/alum i.p. immunization of *Clec9a.Bcl6^KO* and control mice. Foxp3⁻: T follicular helper cells (Tfh) are gated as CXCR5^intPD-1^int, and germinal center Tfh are gated as CXCR5^hiPD-1^hi. Foxp3⁺: T follicular regulatory cells are gated as CXCR5^hiPD-1^hi. **H** Quantification of data shown in (**G**). Statistical analysis was performed by the Mann–Whitney test and exact P values were annotated. Data are pooled from two independent experiments with 4–5 mice per group, and lines represent means ± SD. **I** Experimental design. **J** NP-specific total IgG in the serum from NP-KLH and alum immunized *Clec9a.Bcl6^KO* and control mice at indicated time-points as determined by ELISA for low (NP36)- and high (NP2)-affinity antibodies. Data are representative of two independent experiments with 4–5 mice, and lines represent means ± SED. Two-way ANOVA with Bonferroni post-hoc, exact P values are annotated. Circles: females; triangles: males.

Single-cell suspensions of the intestinal LP were prepared as described previously[60]. Briefly, the intestines were flushed with HBSS supplemented with FCS (10%), Peyer's patches were removed, and epithelial cells and mucus were removed by incubating tissue in HBSS supplemented with FCS (10%) and EDTA (2 mM) 3 times each for 15 min at 37 °C with continual shaking at 450 rpm. The remaining tissue pieces were digested in R10 media (RPMI 1640 supplemented with FCS (10%) containing DNase I grade II (30 μg/ml) and Liberase TM (58 μg/ml) for 20 min at 37 °C with a magnetic stirrer.

Lungs were perfused with PBS prior to resection. All lung lobes were cut into small pieces and digested in R5 media (RPMI 1640 supplemented with FCS (5%) containing DNase I grade II (30 μg/ml) and Liberase TM (50 μg/ml, Roche) for 45 min at 37 °C with a magnetic stirrer.

Intestinal LP and lung cell suspensions were filtered through 100 μm cell strainers (Fisher Scientific), and leukocytes were enriched by density gradient centrifugation with 40%/70% Percoll (GE Healthcare) prior to further analysis.

### Flow cytometry

Single-cell suspensions were stained for multicolor analysis with the indicated fluorochrome- or biotin-conjugated antibodies after blocking with Fc block (purified anti-CD16/32). Antibodies were diluted in flow cytometry buffer (DPBS, 2% FCS), and staining was performed for 30 min on ice in the dark. LIVE/DEAD™ Fixable Near-IR Dead Cell Stain Kit (Invitrogen) or Fixable Viability Dye eFluor™-780 (Invitrogen) were used for determining the viability of cells.

Lymphocytes from *C. rodentium* infection experiments were treated with rmIL-23 (40 ng/ml, R&D) for 3 h at 37 °C with 5% CO₂, washed and further incubated with rmIL-23 (40 ng/ml), PMA (50 ng/ml, Sigma Aldrich), Ionomycin (750 ng/ml, Sigma Aldrich) and BD Golgi-Stop™ (containing monensin at 1:1000 dilution, BD) for an additional 3.5 h at 37 °C with 5% CO₂.

For intracellular staining, cells were fixed with the Foxp3 Fixation/Permeabilization kit from eBioscience. Samples were acquired on a Fortessa X20, an LSRFortessa, or a Symphony flow cytometer (BD Biosciences) using FACSDiva software (BD Biosciences) and analyzed with FlowJo software (Tree Star). All antibodies and working dilution are listed in Suppl. Data 4.

### Splenic dendritic cell sorting and cDNA library preparation for RNA-seq

Splenic DC was enriched using the Dynabeads Mouse DC Enrichment kit (Invitrogen) and sorted on a BD FACS Melody to obtain cDC1 (B220⁻CD64⁻CD3⁻CD19⁻NK1.1⁻CD11c^hiMHCII⁺XCR1⁺) and cDC2 (B220⁻CD64⁻CD3⁻CD19⁻NK1.1⁻CD11c^hiMHCII⁺CD11b⁺XCR1⁻). Sorted cells (2-4×10⁴ per sample) were resuspended in 350 μl RNeasy Lysis Buffer (QIAGEN), and RNA was extracted using the QIAGEN RNeasy

Mini kit (QIAGEN). RNA was washed, treated with DNase I (QIAGEN), and eluted per the manufacturer's instructions. RNA-seq libraries were prepared by combining the Nugen Ovation RNA-seq system V2 with NuGEN's Ultralow System V2 (NuGEN Technologies). The amplified libraries were purified using AMPure beads, quantified by qPCR, and visualized on an Agilent Bioanalyzer using BioA DNA High sensitivity (Agilent). The libraries were pooled equimolarly and run on a HiSeq 2500 as single-end reads of 50 nucleotide length.

### RNA-sequencing data analysis

Sequencing reads were mapped to the mouse reference genome (GRCm38.85/mm10) using the STAR aligner (v2.7.6a)[61]. Alignments were guided by a Gene Transfer Format (Ensembl GTF GRCm38.101). Read count tables were normalized based on their library size factors using DESeq2 (v1.30.1)[62], and differential expression analysis was performed with a filter of log2foldChange > 1 and a *p*-adjust value < 0.01. For selected gene signatures (cDC1 and cDC2 or ESAM^hi and ESAM^lo DC2s), the 100 DEGs with the highest normalized count values were selected, out of which then the top 50 genes with the highest absolute log2FoldChange were depicted in heatmaps. RNA-seq reads aligned to genes from the GENCODE gene annotation were counted using Ensembl's BiomaRt-v2.46.3. Heatmap visualizations were done using pheatmap v1.0.12. Downstream statistical analyses and plot generation were performed in the R environment (v4.0.5)[63]. The online tool Venny 2.1 was used for the drawing of the Venn plots. Package PlotMA was used for visualizing gene expression changes from two different conditions (parameters: alpha = 0.01, type ="ashr")[64]. Principal Component Analysis (PCA) plots were drawn using PCAtools 2.2.0, with removeVar=0.01[65]. Volcano plots were drawn using the function of ggpubr 0.4.0[66]. Gene Set Enrichment Analysis was performed using Gene Ontology R package clusterProfiler 3.18.1 (category biological process) on DEGs obtained from DESeq2. The *p*-values were adjusted using "BH" (Benjamini–Hochberg) correction, and the cutoff was 0.01[67]. GO biological process terms were ranked by GeneRatio. The gene ratio is defined as the number of genes associated with the term in DEGs divided by the number of DEGs. For KEGG enrichment, DESeq2-derived DEGs were used[68]. Enrichments with a *p*-value from Fisher's exact test ≤ 0.05 were considered significant.

### Confocal microscopy

Mice were sacrificed, spleens taken out, immediately washed with cold DPBS, and fixed with either 4% PFA in PBS for 12 h or 1% PFA in PBS for 24 h, followed by an overnight wash with washing buffer PBS-XG (PBS + 5% FCS + 0.2% Triton X-100). Samples were embedded in warm 4% low melting agarose for sectioning. 60-100 μm sections were cut using a Vibratome 1200 S (Leica) and stored until use. For staining, sections were incubated with primary antibodies in 500 μl washing buffer in a 24-well plate overnight at 4 °C under constant agitation

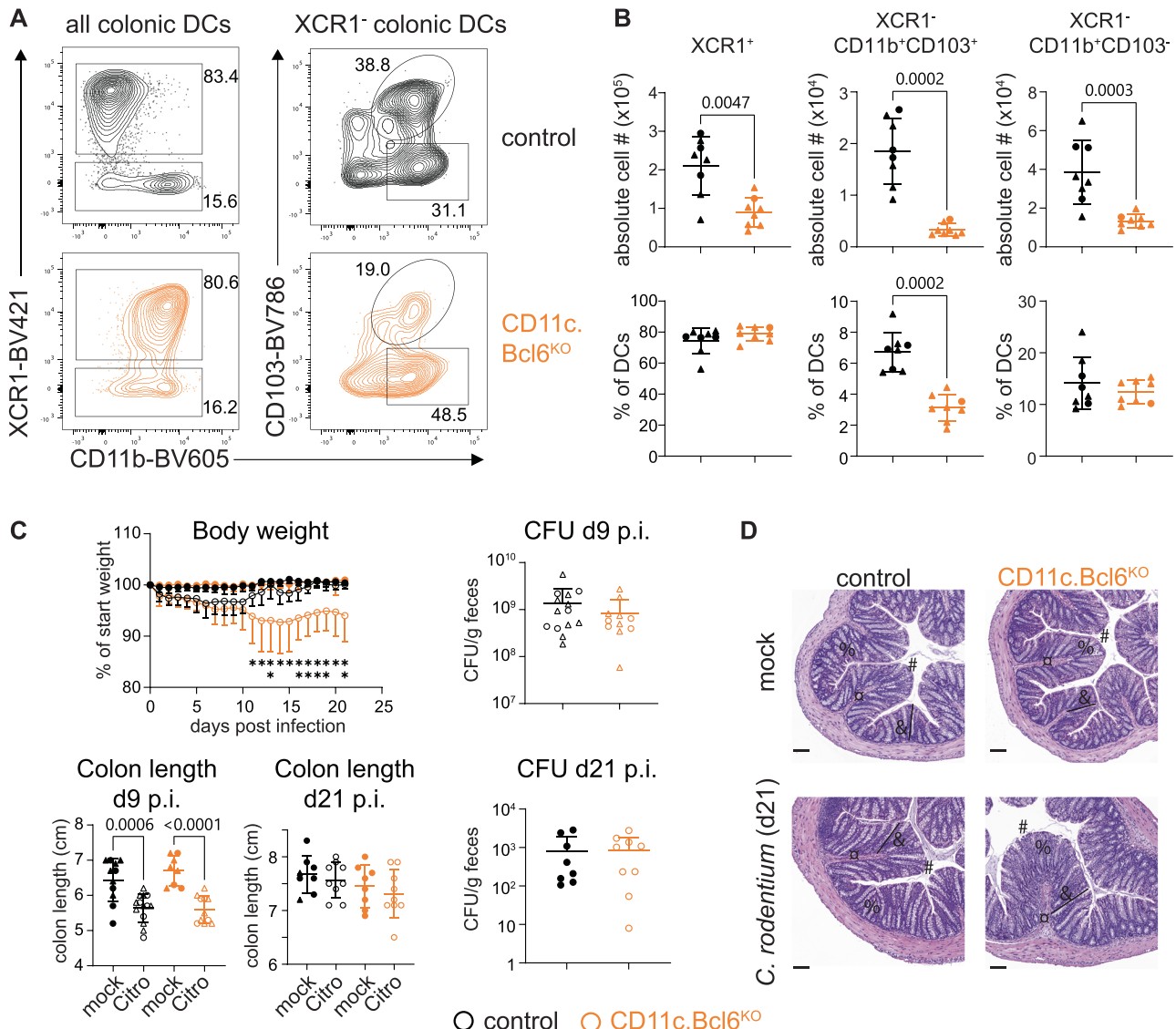

**Fig. 9 | *Citrobacter rodentium* infection in mice with Bcl6 deletion in cDC. A** Left: Flow cytometry plots showing cLP XCR1⁺ and XCR1⁻CD11b⁺ DC subsets gated on all dendritic cells (DC) (live, CD45⁺, lineage⁻ (CD3, CD19, CD64, B220, NK1.1), MHCII⁺, CD11cʰⁱ single cells. Right: XCR1⁻ DC is further subdivided into CD103⁺ and CD103⁻ classical DC2 (cDC2) from control and *CD11c.Bcl6ᴷᴼ* mice. **B** Absolute numbers (top) and proportions (bottom) of colon lamina propria XCR1⁺, XCR1⁻CD103⁺CD11b⁺, and XCR1⁻CD103⁻CD11b⁺ DC subsets from control and *CD11c.Bcl6ᴷᴼ* mice. Data points represent values from individual mice pooled from 3 experiments with 2–3 mice each, and lines represent means ± SD. Statistical analysis using the Mann–Whitney test, exact *P* values are annotated. **C** Percentage of weight loss of mock and *C. rodentium* infected control and *CD11c.Bcl6ᴷᴼ* mice. Each symbol represents the mean weight loss of all mice within the group at the indicated time point post-infection versus their starting weight, and error bars represent SD; statistical analysis using two-way ANOVA (and Tukey's multiple comparison test) with Geisser–Greenhouse correction, *\**P* < 0.05 and \*\**P* < 0.01 (top left). Colon lengths in mock and infected mice at 9 or 21 days post inoculation; statistical analysis by one-way ANOVA (and Tukey's multiple comparison test), exact *P* values are annotated (bottom left). *C. rodentium* titers in feces were measured as colony-forming units (CFU) of infected control and *CD11c.Bcl6ᴷᴼ* mice at indicated days post-infection; statistical analysis using Mann–Whitney test, not significant (ns, right). All data points represent values from individual mice, lines represent means ± SD, and data is pooled from 2 to 3 experiments with 3–5 mice per mock/*C. rodentium*-infected group. **D** H&E-stained sections of the distal colon of mock and *C. rodentium* infected control and *CD11c.Bcl6ᴷᴼ* mice 21 days post-inoculation. Scale bar, 100 μm. Symbols: # epithelial lining; ¤ leukocyte infiltration; % goblet cell destruction; & villus length. Representative of 4–5 mice per group from one experiment.

(120 rpm), followed by at least 6 h of washing in 5 ml washing buffer. To enhance the YFP signal, a primary unlabeled anti-GFP antibody was used in the experiments based on *Clec9a.Bcl6ᵂᵀ* and *Clec9a.Bcl6ᴷᴼ* mice, each carrying homozygous Rosa26fl-Stop-fl-YFP alleles. This antibody was detected in a secondary antibody incubation step overnight in 500 μl PBS-XG in a 24-well plate, followed by an additional washing step in 5 ml washing buffer for 6 h. All antibodies and working dilution are listed in Suppl. Data 4. Sections were mounted using ProLong Gold Glass and images were acquired on a Zeiss LSM710 or a Zeiss LSM900 confocal laser microscope. Digital pictures were

acquired with Zen software (Zeiss) and processed using Imaris Viewer (Oxford Instruments).

**Cross-presentation assays**

In vivo: Naïve CD8⁺ T cells from OT-I donors were enriched using the Dynabeads® Untouched™ Mouse T Cells kit (Invitrogen) and stained in 1 ml DPBS with CellTrace Violet dye (Invitrogen) for 10 min at 37 °C. 1×10⁶ OT-I cells were transferred into recipients one day before i.p. injection of 5×10⁶ cells/mouse of H-2^βm1 MEFs expressing truncated non-secreted OVA (OVA-MEF)[17]. One day after immunization,

**A** day 9 post infection

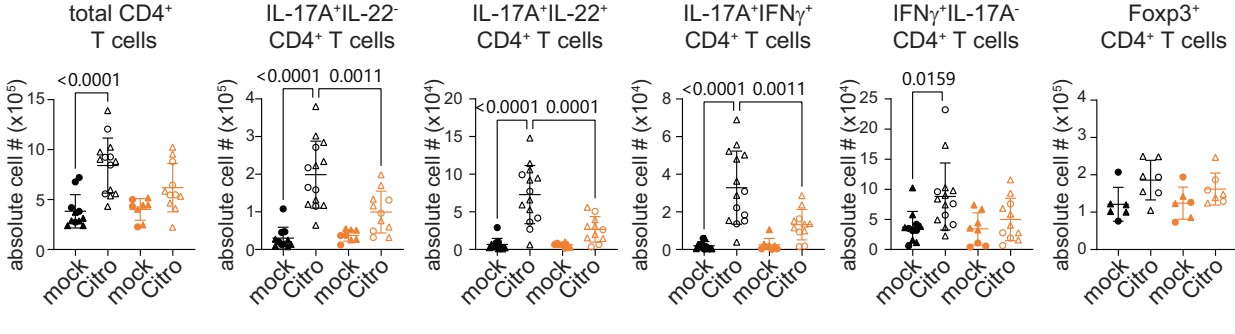

○ control    ○ CD11c.Bcl6^KO

**B** day 21 post infection

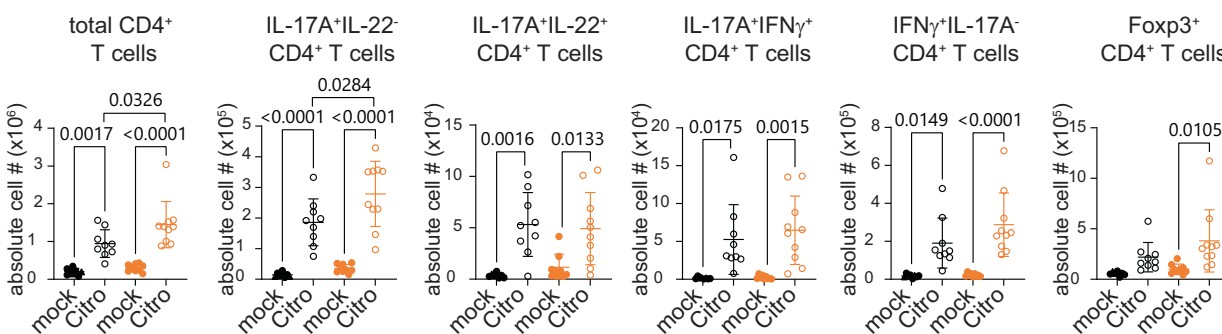

**Fig. 10 | T cell responses to *Citrobacter rodentium* infection in mice with Bcl6 deletion in cDC. A** Numbers of total CD4⁺ T cells, IL-17A⁺IL-22⁻, IL-17A⁺IL-22⁺, IL-17A⁺IFNγ⁺, IFNγ⁺IL-17⁻ and Foxp3⁺ CD4⁺ T cells in the cLP of mock and *C. rodentium* infected control and *CD11c.Bcl6^KO* mice at day 9 post-inoculation. Data points represent values from individual mice pooled from 2 to 4 experiments with 2–4 mice per group, and lines represent means ± SD. Statistical analysis is performed by one-way ANOVA, and exact *P* values are annotated. **B** Numbers of total CD4⁺ T cells,

IL-17A⁺IL-22⁻, IL-17A⁺IL-22⁺, IL-17A⁺IFNγ⁺, IFNγ⁺IL-17⁻ and Foxp3⁺ T cells in the cLP of mock and *C. rodentium* infected control and *CD11c.Bcl6^KO* mice at day 21 post-inoculation. Data points represent values from individual mice pooled from 2 experiments with 4–5 mice per group, and lines represent means ± SD. Statistical analysis is performed by one-way ANOVA, and exact *P* values are annotated. Circles: females; triangles: males.

recipients were injected with FTY720 (20 μg/mouse, Cayman) in saline i.p. Mice were sacrificed three days after immunization.

*In vitro:* DCs were enriched from spleen and mLN using the Dynabeads Mouse DC Enrichment kit (Invitrogen), and naïve CD8 T cells from OT-I donors were isolated using the Dynabeads Untouched Mouse T Cells kit (Invitrogen), both according to manufactures protocol. OT-I cells were stained with CellTrace Violet dye (1:1000, Invitrogen) for 10 min at 37 °C. Enriched DCs from both mLN and spleen were incubated with OVA (EndoFit, InvivoGen) at a concentration of 250 μg/ml in culture media (RPMI 1640 supplemented with 10% FCS, 55 μM β-mercaptoethanol and Penicillin-Streptomycin-Glutamin 100× (Gibco)) for 2 h at 37 °C and washed twice. DCs for negative control were not incubated with OVA. The frequency of cDC1s in each sample was determined using flow cytometry, and 670 cDC1s were plated in a cell culture treated 96-well U bottom plate with $0.2 \times 10^6$ OT-I cells in culture media supplemented with 1 μg/ml LPS. For the positive control, $0.2 \times 10^6$ OT-I cells were stimulated with 10 μg/ml aCD3 (eBioscience, clone 17A2) (plate bound) and 1 μg/ml aCD28 (eBioscience, clone 37.51). Flow cytometry-based analysis was carried out after incubation for 65 h at 37 °C with 5% $CO_2$.

### Induction of Tfh cells and antibody responses in vivo
For adoptive transfer experiments into *CD11c.Bcl6^KO* or control hosts, naïve CD4⁺ T cells were enriched from homozygous TCR-transgenic OT-II (CD45.1/.2) donors using the Dynabeads Untouched™ Mouse CD4 Cells kit (Invitrogen). $1 \times 10^5$ OT-II cells were transferred intravenously (i.v.), and one day later, recipients were immunized i.p. with

ovalbumin grade V (OVA, 0.5 mg, Sigma Aldrich) together with poly(I:C) (100 μg, Sigma-Aldrich). Controls received PBS alone, and mice were sacrificed three days after immunization. For adoptive transfer experiments into *Clec9a.Bcl6^WT* or *Clec9a.Bcl6^KO* hosts, naïve CD8⁻CD19⁻CD44^int/low^CD62L^hi^CD25⁻ CD4⁺ T cells were sorted from heterozygous OT-II (CD45.1/2) donors on BD FACSAria III or Fusion cell sorters. $5 \times 10^{54}$ OT-II cells were injected i.v., and one day later, recipient mice were immunized i.p. with a 1:1 mixture of 50 μg NP₁₇-OVA (Biosearch Technologies/BioCat) in PBS and Imject Alum (Thermo Fisher). Spleens were dissected and analyzed by flow cytometry on day 7. For assessment of endogenous Tfh and Tfr cell responses, *Clec9a.Bcl6^WT* or *Clec9a.Bcl6^KO* mice were immunized i.p. with 50 μg NP₃₂-KLH (Biosearch Technologies/BioCat) in PBS and Imject Alum (Thermo Fisher). Spleens were dissected and analyzed by flow cytometry on day 14. In independent NP₃₂-KLH/alum-immunized cohorts, blood samples were collected on days 7, 14, 21, and 28 using Microvette 200 Serum Gel CAT tubes (Sarstedt), and NP-specific serum antibody levels were determined by ELISA.

### ELISA
IL-6 levels in the serum of mice injected intraperitoneally with PBS or 100 μg poly(I:C) 2 h before analysis were determined using BD OptEIA™ Mouse IL-6 ELISA Set.

To measure NP-specific serum IgG1 and total IgG, half-area radio-immunoassay plates (#3690, Corning) were coated with 25 μl of 10 μg/ml NP-BSA in PBS (N-5050XL-10-BS with loading ratio of 2 or N-5050H-10-BS with loading ratio of 36, both Biosearch Technologies/BioCat)

overnight at 4 °C. The next day, plates were washed four times with PBS + 0,005% Tween20 and then blocked with 70 µl PBS + 2% BSA for 2 h at RT. After four washing steps, serial dilutions of serum from immunized mice were added to the plate and incubated on a plate shaker for 2 h at RT. Pooled sera from day 28-immunized mice were used as standard. Next, plates were washed four times, and 25 µl of a 1:500 dilution of the respective detection antibody (goat anti-mouse IgG(H + L)-alkaline phosphatase (AP) conjugate (Southern Biotech/Biozol) or goat anti-mouse IgG1-AP (Southern Biotech/Biozol) was added to each well and incubated for 45 min while shaking. Plates were washed four times and then incubated with 70 µl of Alkaline Phosphatase Solution (Sigma Aldrich) for 20 min. Plates were read for absorbance at 405 nm (TriStar plate reader). Resulting data were plotted as concentration ($x$-axis) versus OD (OD, $y$-axis) in Excel, then fit to a sigmoidal 4-parameter logistic (4PL) model by utilizing the four parameter logistic (4PL) curve calculator (AAT Bioquest Inc.). The sigmoidal 4PL equations were then used to calculate the concentration of individual samples. Values are expressed as arbitrary units (AU).

### *Citrobacter rodentium* infection

The *C. rodentium* strain DBS100 (ATCC 51459; American Type Culture Collection) was used as described[69] with some amendments. Briefly, *C. rodentium* cultures were grown in Luria-Bertani (LB) medium to an $OD_{600}$ of 1–1.5. Mice were orally inoculated with LB broth or $2 \times 10^9$ CFU *C. rodentium*, and their weight was monitored daily. Fecal samples were collected on indicated days post-infection, serially diluted in PBS and plated on Brilliance™ *E. coli*/coliform Agar (Thermo Fisher). *C. rodentium* colonies, from at least three dilutions per sample, were enumerated and normalized to the weight of feces.

For histopathology, 0.5 cm of the distal colon was immediately placed in 10% formalin and processed by the DTU Histology Core for hematoxylin and eosin (H&E) staining. Images were acquired on a Pannoramic MIDI II scanner (3DHISTECH) and visualized with Case-Viewer software 2.4 (3DHISTECH). The severity of the infection was assessed in a blinded manner.

### Statistical analysis

Statistical analyses were performed with Graphpad Prism 9.0 (Graphpad Software). While symbols in data graphs give information on the sex of the individual mice in the experiments, we did not perform sex-stratified statistics due to variable group size.

### Reporting summary

Further information on research design is available in the Nature Portfolio Reporting Summary linked to this article.

## Data availability

The RNA-sequencing data generated in this study have been deposited in the NCBI SRA database under accession code PRJNA834905. Flow cytometry data is available upon request. Source data are provided in this paper.

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

## Acknowledgements

We thank Katrine Fog Starup, Julien Vandamme, Frank Dahlström, and Jennifer-Christin Becker for excellent technical support, the Lahl, Agace, Bekiaris, and Baumjohann laboratories for fruitful discussions, and Prof Allan Mowat for intellectual input and manuscript editing. *Bcl6^{fl/fl}* mice were kindly gifted by Alexander Dent, and the OVA-MEF line was from Caetano Reis e Sousa. H&E sections were processed by Susanne Primdahl, and histology data was generated using research infrastructure at DTU National Food Institute, including FOODHAY (Food and Health Open Innovation Laboratory, Danish Roadmap for Research Infrastructure). We acknowledge the BMC Core Facility Flow Cytometry of LMU Munich for providing equipment and the Core Facility of the Medical Faculty at the University of Bonn for providing support and instrumentation funded by the Deutsche Forschungsgemeinschaft (DFG, German Research Foundation) (387333827, 216372545, 216372401, 387335189).

Funding for this project was provided by the China Scholarship Council fellowship (HX), the Lundbeck Foundation Fellowship R215-2015-4100 and R396-2022-373 (K.L.), the Ragnar Söderberg Foundation Fellowship in Medicine (K.L.), Vetenskapsrådet 2014-3595 (K.L.), Cancerfonden 21 1826 Pj (K.L.), the Novo Nordisk Foundation 18038 (K.L.), the Crafoord Foundation (K.L.), the European Research Council ERC-2016-STG-715182 (B.S.), Deutsche Forschungsgemeinschaft (DFG, German Research Foundation) Emmy Noether grant Schr 1444/1-1, Project-ID 360372040—SFB 1335 (project 8) and 322359157-FOR2599-A03 (B.S.), Deutsche Forschungsgemeinschaft (DFG, German Research Foundation) Emmy Noether Program BA 5132/1-1 and BA 5132/1-2 (252623821), SFB 1054 Project B12 (210592381) and Germany's Excellence Strategy EXC2151 (390873048) (D.B.).

## Author contributions

Conceptualization: D.B. and K.L. Methodology: H.X., I.U., J.H., H.N., L.B., I.K., Y.C., S.H., U.M., L.Z., A.S., C.-F.P., B.S., D.B. and K.L. Investigation: H.X., I.U., J.H., H.N., L.B., I.K., S.H., U.M., L.Z., B.S., D.B. and K.L. Visualization: H.X., I.U., U.M., D.B. and K.L. Funding acquisition: H.X., D.B., B.S. and K.L. Project administration: K.L. Supervision: C.-F.P., A.S., B.S., D.B. and K.L. Writing—original draft: K.L. Writing—review & editing: I.U., B.S., D.B. and K.L.

## Competing interests

The authors declare no competing interests.
