## [Peer Review File · Nature Communications]

Genomic deletion of Bcl6 differentially affects conventional dendritic cell subsets and compromises Tfh/Tfr/Th17 cell responsesREVIEWER COMMENTS

Reviewer #1 (Remarks to the Author):

In the manuscript "Loss of Bcl6 transcriptionally alters classical dendritic cells and diminishes T follicular helper cell and Th17-inducing cDC2 in mice" by Hongkui Xiao et al., the authors revisited the role of Bcl6 in mouse cDC1. Using XCR1cre, CD11Ccre, Clec9acre-Bcl6fl/fl mice, the authors found that, in contrast to their 2014 paper [PMID: 24292363], Bcl6 is dispensable for the development and cross-presentation function of mouse splenic cDC1. The results, thus, are in line with the 2021 publication by Murphy KM's group [PMID: 34135058].

The authors then found that the CD11Ccre and Clec9acre-Bcl6fl/fl mice lack the ESAMhi cDC2 population. And further proposed that Bcl6 expression in cDC2 is important for Tfh cell induction and Th17 promotion. The strength of the manuscript lies in the multiple cre strains used, which improves the rigor of the study. The weakness is in the Tfh and Th17 study. The Tfh study is preliminary and not convincing. The interpretation of the Th17 study needs to be precise since only CD11CcreBcl6fl/fl mice were used. In all, the current data does not support the conclusion that loss of Bcl6 decreases Tfh cells and Th17-inducing cDC2. Below are my comments.

Major points:

1. Figure 7D-7F examining splenic Tfh cell induction by Poly(I:C)/OVA and Alum/OVA are not convincing.
 - What was the justification for the adoptive transfer of OT-II cells? Poly(I:C)/OVA or Alum/OVA (i.p) should generate OVA-specific spleen Tfh cells in the cre-negative Bcl6fl/fl mice without the transfer OT-II cells.
 - Why chose day 3 for Figure 7E and day 7 for Figure 7F for Tfh detection? Tfh cells usually peak at day 14 after immunization. For rigor, please show the time course of Tfh to justify the selection of these timepoints.
 - The authors need to do FoxP3 stain to separate Tfh vs Tfr. Both are Bcl6+ CXCR5+PD1+.
 - The Tfh flow data in Fig7E and 7F are not convincing. What was the rationale for using Bcl6-CXCR5 in Fig 7E rather than PD-1/CXCR5 to identify Tfh as in Fig 7F? CXCR5 expression depends on Bcl6.
 - To improve rigor and demonstrate the change in Tfh cells is biologically significant, the authors need to show sera anti-OVA IgG1, IgG2A/2C to demonstrate the Tfh difference they observed alter the antibody production in both titers and affinity.
2. Figure 6, though the lack of splenic ESAMhi cDC2 is clear in the CD11Ccre and Clec9acre-Bcl6 mice, it is not clear whether it is due to the ESAMhi cDC2-intrinsic expression of Bcl6. The Clec9acre deletes Bcl6 in cDC1 and, to a less degree cDC2, pDCs. The interpretation needs to be precise.
3. Figure 8 showed that CD11CcreBcl6fl/fl mice had impaired Th17 responses during C. rodentium infection. The authors contributed to the Bcl6 expression in cDC2. Such an interpretation is not accurate. CD11Ccre will delete Bcl6 in all CD11c cells, including some B cells, cDC1, cDC2, and moDCs. The authors need to repeat the experiment with Clec9acre mice to exclude CD11C+ B cells and moDCs. And better, do an adoptive transfer experiment to put back WT ESAMhi cDC2 into the CD11Ccre mice and see if it restored the Th17 responses.

Minor points:

- Figure 5, The defective OVA-MEF responses in mLN is worth further investigation. It will be more convincing to use the LCMV model and examine Granzyme B, CTL assay to confirm the lack of CTL activity in mLNs of XCR1creBcl6fl/fl mice.
- For OVA immunization, the authors used ovalbumin grade V (OVA, Sigma Aldrich) and poly(I:C) (Sigma-Aldrich) that likely contain endotoxin.

Reviewer #2 (Remarks to the Author):

This study by Xiao et al. performed comprehensive analysis of three distinct conditional Bcl6-deficient mouse models for conventional dendritic cell phenotype, transcriptome and function. The authors showed that deletion of Bcl6 altered the phenotype and transcriptome of cDC1 and cDC2, but their lineage identity is preserved. Functionally, the authors found the Bcl6-deficient mice

maintained their ability to cross-present antigen to CD8+ T cells, but induced fewer T follicular helper cells in response to the model antigen ovalbumin and mounted diminished TH17 immunity to *Citrobacter rodentium* in the colon. While this study is overall well performed and presented, two major concerns need to be addressed before it can be considered for publication.

1. The biggest concern about this study is the claim of "Bcl6 deficiency diminished cDC1 in the periphery, and Notch2-dependent cDC2 in the spleen and intestinal lamina propria". As the authors have discussed in their introduction, a previous study showed that "down-regulation of CD11c in the absence of Bcl6 may have compromised the detection of" cDCs. In current study, the authors still used CD11c to define classical DCs, it is unclear whether Bcl6 deficiency diminished periphery cDC1 and Notch2-dependent cDC2, or Bcl6 deficiency only down-regulated CD11c in those cDC subsets. An alternative method that is independent of CD11c expression is needed to draw this important conclusion, for example using Zbtb46 expression to define classical DCs. More specifically:

1) In Figure 1, for splenic and resident LN cDC analysis, could the authors show their gating strategy for cDCs?

2) In Figure 2, for migratory LN cDC and small intestine lamina propria cDC analysis, could the authors show their gating strategy for cDCs? An alternative method for defining cDC besides pre-gate on CD11c is needed here to distinguish cDC deficiency with CD11c down-regulation.

3) In Figure 6, for panels A and C, could the authors explain in the figure legend the pre-gating strategy of the splenocytes, or whether the splenocytes underwent DC enrichment? Why is there only CD11c positive cells? As the authors discussed in their result section, "the ESAMlo subset of cDC2 expressed slightly lower levels of CD11c compared to their ESAMhi counterparts". Due to the difference in CD11c level between ESAMhi and ESAMlo cDC2 subsets, an alternative method for defining cDC besides pre-gate on CD11c is required to draw the conclusion that "Bcl6 deficiency in DC causes the loss of the ESAMhi lineage of splenic cDC2".

4) In Figure 7C, the authors still used CD11c as the pan-DC marker, loss of CD11c positive cells in microscopy can be due to the loss of DC or down-regulation of CD11c expression. An alternative DC marker is required to draw the conclusion of "confirmed a substantial loss of DC from this region in the CD11c.Bcl6KO spleen".

2. The authors discovered a surprising finding that XCR1+ cDC that preserved in Bcl6-deficient mice express the cDC2 marker CD11b. It would be important to determine whether this reflects acquisition of CD11b by cDC1, or cDC2 with aberrant XCR1 expression. But the strategy chosen by the authors raises some concern. To answer this question, could the authors sort XCR1+CD11b- cDC, XCR1+CD11b+ cDC and XCR1-CD11b+ cDC separately from CD11c.Bcl6KO mice, and compare their transcriptional profile with XCR1+CD11b- cDC and XCR1-CD11b+ cDC from WT mice. The XCR1 positivity should be determined by XCR1 level in WT mice. Principal component analysis of these datasets would answer the above question in a more unbiased manner. With current data, it is not clear whether the XCR1+ cDC (independent of CD11b expression) are cDC1 with aberrant CD11b expression, or are a combination of cDC1 with a third XCR1+CD11b+ cDC population.

Minor point:

In Figure 8C, could the two panels of colon length be mis-labelled?

Reviewer #3 (Remarks to the Author):

The manuscript by Hongkui Xiao et al. titled "Loss of Bcl6 transcriptionally alters classical dendritic cells and diminishes T follicular helper cell and Th17-inducing cDC2 in mice" describes Bcl6-deficiency in the DC modifies the phenotype as well as the gene expression of cDC1 and cDC2. There are controversial data regarding the role of TF Bcl6 in all cDC, and these findings point to a relevant impact of Bcl6-deficiency in the immunological function of cDC1 and cDC2. The authors have done an exciting study addressing the impact of Bcl6 deletion in murine cDC, but additional comments and concerns are raised below.

Major points.

Figure 1 and Figure 2. The absence of Bcl6 results in exciting findings that cDC1 can acquire some cDC2 markers. The same double-positive cells were observed in the LNs and SI lamina propria. Since the XCR1+CD11b+ cDC are so evident in different compartments, it would be interesting to

include a specific flow cytometry analysis in XCR1/CD11b double positive cells, delimiting the gate only in XCR1+/CD11b+ (figure 1 A). Also, it will be very informative to include the frequency (%) inside of the gate and also a graph showing the MFI of CD11b comparing all groups. Indeed, the expression of CD11b (MFI) in CD11c.Bcl6Ko is clearly more pronounced in comparison to XCR1.Bcl6Ko, and in the periphery and mesenteric resident LN most of the DCs of XCR1+ shift to the CD11b+ population.

Figure 4 A. It would be interesting to include the MFI of all groups in the representative histograms because there was a slight shift in some molecules, for example, TLR3 and IRF8, compared to the control. Moreover, the data points graph pooled from two independent experiments could be included in this Figure 4 A, in addition to the representative histogram.

The figure 4B shows that the absence of Bcl6 modulates some relevant gene ontology pathways that could modulate some immunological functions of XCR1+ cDC, for example, the production of IL-1, regulation of cell adhesion, and response to the virus. Have you considered whether the XCR1/CD11b double-positive cells acquire gain from one of these immunological functions? For example, splenic DCs can secrete substantial amounts of mature IL-1 β upon stimulation with TLR ligands, such as TLR3, that were upregulated on splenic XCR1.Bcl6KO. Could XCR1/CD11b double-positive cells express more Nlrp3 and produce a higher amount of IL-1 β ? Even though the results from Figure 6 show interesting findings that the absence of Bcl6 does not affect the capacity to cross-present antigen, it would be interesting to evaluate if so; other immunological functions could be affected, such as in virus infection or in the presence of TLR ligands.

Pg7, line 197. The author described, "OTI proliferation was also observed in the mLN. In the mLN of XCR1.Bcl6KO mice OTI cells showed somewhat diminished proliferation compared to control mice; however, this most likely reflects the decreased number of migratory cDC1 in the mLN of XCR1.Bcl6KO mice (Fig. 2A)." I agree that the impairment of migration cDC observed previously could explain the lower proliferation of OT-I. However, to confirm if these cells preserve the ability to cross-present antigen to OT-I, instead of transferring OT-I into recipients, it would be interesting to sort cDC1 from XCR1.Bcl6KO mice were previously immunized with heat-shocked OVA-MEFs and co-culture in vitro with OT-I cells to evaluate cell proliferation.

Pag. 8, line 210. The authors described, "The remaining cDC in the spleen of CD11c.Bcl6KO mice also showed somewhat lower expression of MHCII and higher expression of CD11b than control cDC2 (Fig. 6A)." As observed in figure 6A, splenic cDC from CD11c.Bcl6KO mice express higher CD11b than control; however, the percentage of MHC-II on these cells was similar in both groups, control (96.4%) and CD11c.Bcl6KO (92.6%). It seems like the MFI of MHC-II is lower in CD11c.Bcl6KO than the control. However, there needs to be more information about the MFI in this figure. I suggest including the data points graph pooled from independent experiments in addition to the representative dot and contour plots.

Figure 8. The deficiency of Bcl6 affects the Th17 priming at day 9; however, after 21 days, most parameters were recovered (colon length, IL-17+IL-22; IL-17+/IFN-g; IFN-g+IL-17- cells). Even though the Th17 differentiation was affected initially, in the discussion, the authors described, "We cannot currently explain the differences in the early innate immune response between CD11c.Notch 2KO and CD11c.Bcl6KO mice, but our data are consistent with the idea that DC play the dominant role in coordinating the adaptive immunity required to terminate the infection. However, while Bcl6-dependent cDC2 support optimal Th17 induction to *C. rodentium*, they are not essential for this response, as Th17 numbers normalized over time." Since we observed a slight increase in the IL-17+ L-22; IL-17+/IFN-g cell number after 9 days, have you considered that Th17 cell expansion in infected CD11c.Bcl6ko mice, could explain the cell numbers normalized over time (after 21 days)?

Minor points

Pg4, line 110. Be consistent in describing XCR1-CD103+CD11b+ population. In Figure 2, the graph was indicated as XCR1-CD11b+ CD103+.

Reviewer #1 (Remarks to the Author):

In the manuscript “Loss of Bcl6 transcriptionally alters classical dendritic cells and diminishes T follicular helper cell and Th17-inducing cDC2 in mice” by Hongkui Xiao et al., the authors revisited the role of Bcl6 in mouse cDC1. Using XCR1cre, CD11Ccre, Clec9acre-Bcl6fl/fl mice, the authors found that, in contrast to their 2014 paper [PMID: 24292363], Bcl6 is dispensable for the development and cross-presentation function of mouse splenic cDC1. The results, thus, are in line with the 2021 publication by Murphy KM’s group [PMID: 34135058].

The authors then found that the CD11Ccre and Clec9acre-Bcl6fl/fl mice lack the ESAMhi cDC2 population. And further proposed that Bcl6 expression in cDC2 is important for Tfh cell induction and Th17 promotion. The strength of the manuscript lies in the multiple cre strains used, which improves the rigor of the study. The weakness is in the Tfh and Th17 study. The Tfh study is preliminary and not convincing. The interpretation of the Th17 study needs to be precise since only CD11CcreBcl6fl/fl mice were used. In all, the current data does not support the conclusion that loss of Bcl6 decreases Tfh cells and Th17-inducing cDC2. Below are my comments.

We thank the reviewer for the valuable feedback. In response, we added several new datasets to our study, with a specific emphasis on better dissecting the role of Bcl6 in DCs with respect to their role in supporting Tfh cell responses in the spleen. We believe these additional data improve our manuscript and are grateful for the given suggestions.

Major points:

1. Figure 7D-7F examining splenic Tfh cell induction by Poly(I:C)/OVA and Alum/OVA are not convincing.

- What was the justification for the adoptive transfer of OT-II cells? Poly(I:C)/OVA or Alum/OVA (i.p) should generate OVA-specific spleen Tfh cells in the cre-negative Bcl6fl/fl mice without the transfer OT-II cells.

We agree with the reviewer that the different immunizations would also induce endogenous OVA-specific Tfh cells without the need of OT-II transfers. The rationale for using OT-II cells was to test the same antigen-specific wildtype TCR-tg T cells in the different hosts that either lacked Bcl6 in DCs or were Bcl6-sufficient. While Clec9a.cre and CD11c.cre drivers generally do not target T cells, focusing on wildtype TCR-tg cells completely ruled out any potential T cell-intrinsic defects of the responding OT-II cells. Another advantage of this approach is the ability to standardize the initial antigen-specific OT-II cell precursor frequency and to precisely track and quantify the OT-II cells and their Tfh cell phenotype over time (e.g. Baumjohann et al, *Immunity* 2013, PMID: 23499493). To nevertheless accommodate for this reviewer’s comment, we now also include experiments without OT-II transfers, in which we immunized mice i.p. with NP-KLH/alum and assessed endogenous Tfh cell responses 14 days after immunization. Importantly, these experiments further support our initial claim that DC-specific Bcl6 expression is important for Tfh cell induction upon immunization (see revised Figure 8 and Suppl. Fig. 8).

- Why chose day 3 for Figure 7E and day 7 for Figure 7F for Tfh detection? Tfh cells usually peak at day 14 after immunization. For rigor, please show the time course of Tfh to justify the selection of these timepoints.

Tfh cell differentiation is a multi-step process that involves consecutive interactions with antigen-presenting cells, i.e. DCs and B cells, in distinct anatomical sites, ranging from the T zone to the T/B zone border and to the germinal center (GC) in the B cell follicle (Baumjohann and Fazilleau, *Eur J Immunol* 2021, PMID 33788271). In our study, we first addressed the very early steps of Tfh cell differentiation at day 3.5 after immunization, a time frame in which Tfh cell differentiation is mediated by DCs and still independent of cognate interactions with B cells (Baumjohann et al., *J Immunol* 2011, PMID 21804014; Goenka et al., *J Immunol* 2011, PMID 21715693). At day 7, GCs are generally established and fully mature Tfh cells are present as well, thus our second analysis time point was at this stage. When a GC reaches its peak and how long it lasts, depends on the antigen/adjuvant or infection context, and thus it is not possible to determine a universal time point when Tfh cells reach their peak. In fact, in an acute viral infection, e.g. LCMV Armstrong, the virus is cleared after one week so that the peak of the GC response does not extend to day 14. To accommodate for the reviewer's suggestion though, we have added a day 14 time point to our analyses and could show that Tfh cell frequencies are reduced at these late stages as well. For these experiments, we used NP-KLH/alum as inducing agent and quantified the endogenous Tfh cell response (no OT-II). In line with the next comment by the reviewer, we also discovered that Tfr cells were particularly affected by DC-specific Bcl6-deficiency. These new datasets are shown in Figure 8 and in the new Suppl. Figure 8.

- The authors need to do FoxP3 stain to separate Tfh vs Tfr. Both are Bcl6+ CXCR5+PD1+.

We thank the reviewer for pointing out Tfr cells. In the context of our initial OT-II TCR-tg cell adoptive transfer and immunization experiments, no Tfr cells are generated from the naive precursors, as Tfr cells are mainly derived from thymus-derived Tregs. The attached figure shows OT-II cells and endogenous CD4⁺ T cells 3.5 days post NP-OVA/alum immunization, clearly showing that no Treg nor Tfr cells are present in the responding OT-II cells, while there is a reduction in the endogenous pool.

To better address this reviewer's point, we now added new experimental data assessing Tfr cells in the absence of OTII transfer in response to NP-KLH/alum immunization (Figure 8 and Suppl. Figure 8). Indeed, Tfr cells were strongly affected by the absence of Bcl6 in DCs. Such a lack of inhibitory Tfr cells may actually mask the net effects of DC-restricted Bcl6 deficiency on intrinsic Tfh cell responses. This could explain the fact that we did not observe a clear defect in terms of GC Tfh cell frequencies at day 14, in contrast to the day 7 OT-II data. Taken together, our additional experiments extend the impact of Bcl6 in DCs on Tfh cells to Tfr cells as well.

- The Tfh flow data in Fig7E and 7F are not convincing. What was the rationale for using Bcl6-CXCR5 in Fig 7E rather than PD-1/CXCR5 to identify Tfh as in Fig 7F? CXCR5 expression depends on Bcl6.

The combinations of CXCR5/PD-1 and CXCR5/Bcl6 are both widely accepted stainings for the identification of Tfh cells (Eisenbarth et al., 2021 *Trends Immunol*, PMID 34244056), with Bcl6 requiring an extra intracellular transcription factor staining step. The reason for the different usage was solely that the two

experiments were performed in two different laboratories. As we have added new data to the revised manuscript, we now include both combinations in these new experiments for OT-II analysis on day 7 and endogenous CD4⁺ T cell analysis on day 14 after immunization (CXCR5 and PD-1 in the main Figure 8, Bcl6 and PD-1 in Suppl. Fig. 8), further substantiating that Bcl6-deficiency results in impaired Tfh cell responses as assessed by both CXCR5/PD-1 and CXCR5/Bcl6 gating strategies to identify Tfh and Tfr cells.

- To improve rigor and demonstrate the change in Tfh cells is biologically significant, the authors need to show sera anti-OVA IgG1, IgG2A/2C to demonstrate the Tfh difference they observed alter the antibody production in both titers and affinity.

We agree with the reviewer that it is important to also look at the antibody response. To this end, we immunized control and DC-specific Bcl6-deficient mice with NP-KLH/alum and collected serum on days 7, 14, 21, and 28 after immunization for assessment of OVA-specific antibodies as well as an estimate of affinity maturation by ELISA. Since CD11c is expressed by GC B cells and plasma cells (Baumjohann et al, *Immunity* 2013, PMID: 23499493; Hebel et al., *Eur J Immunol* 2006, PMID: 17051619), we performed these assays only in *Clec9a.Bcl6^{KO}* mice and not in *CD11c.Bcl6^{KO}* mice. We found a small, but consistent, defect in NP-specific total IgG antibody levels, as well as for the type-2, alum-based signature isotype IgG1 (Fig. 8J and Suppl. Fig. 8D). Affinity, as determined by increasing ratios of NP2 vs. NP36-binding antibody titers, was also affected by Bcl6 deficiency. The reason for why the difference is not bigger is potentially found in the strong effect on Tfr in addition to Tfh cells in the absence of Bcl6 in DCs (Fig. 8G and H). We added a paragraph to the discussion addressing this point (starting page 15, line 415).

2. Figure 6, though the lack of splenic ESAM^{hi} cDC2 is clear in the CD11Ccre and *Clec9acre-Bcl6* mice, it is not clear whether it is due to the ESAM^{hi} cDC2-intrinsic expression of Bcl6. The *Clec9acre* deletes Bcl6 in cDC1 and, to a less degree cDC2, pDCs. The interpretation needs to be precise.

We agree that the phenotype in cDC2 is surprisingly clear. We however argue that the effect is cell intrinsic given that the cDC2 population is completely preserved in *XCR1.Bcl6^{KO}* mice, specifically targeting cDC1 (see figure, representative for 2 experiments with 3 mice per group). pDCs are poorly targeted using the *Clec9a.cre* driver (Schraml, *Cell* 2013, doi: 10.1016/j.cell.2013.07.014) and their abundance is not affected

by absence of Bcl6 (Ohtsuka, *Jl* 2011, doi: 10.4049/jimmunol.0903714). We realize that this is not proving cell-intrinsic actions by Bcl6 in cDC2, but also do not claim this to be the case: “These data confirm that Bcl6 deficiency in DC causes the loss of the ESAM^{hi} lineage of splenic cDC2” (page 12, line 312). We edited the last paragraph in the discussion accordingly, which now reads: “These data add to the emerging and poorly understood heterogeneity within the cDC2 compartment by establishing that certain cDC2 subsets require Bcl6-

expression in the DC compartment in addition to Notch2- and lymphotoxin β -signaling” (page 16, line 450).

3. Figure 8 showed that CD11CcreBcl6fl/fl mice had impaired Th17 responses during *C. rodentium* infection. The authors contributed to the Bcl6 expression in cDC2. Such an interpretation is not accurate. CD11Ccre will delete Bcl6 in all CD11c cells, including some B cells, cDC1, cDC2, and moDCs. The authors need to repeat the experiment with Clec9acre mice to exclude CD11C+ B cells and moDCs. And better, do an adoptive transfer experiment to put back WT ESAMhi cDC2 into the CD11Ccre mice and see if it restored the Th17 responses.

We have now performed the experiment in *Clec9a.Bcl6^{KO}* mice. While there is a trend towards less Th17, the results are not significant. We found that the targeting efficiency of cDC2 in the gut is generally much lower than what we observe in the spleen, where almost all cDC2 are labelled using the Clec9a driver together with a fate reporter. This finding explains the lack of a significant effect on Th17, but also means that we cannot exclude an effect of Bcl6 in other cells. These data are now included in the manuscript as new Supplementary Figure 9. We also changed the text from “DCs” to “CD11c⁺ mononuclear phagocytes” (page 13, line 341).

Minor points:

- Figure 5, The defective OVA-MEF responses in mLN is worth further investigation. It will be more convincing to use the LCMV model and examine Granzyme B, CTL assay to confirm the lack of CTL activity in mLNs of XCR1creBcl6fl/fl mice.

We agree with the reviewer that this is an interesting future direction. Unfortunately, we do not have an ongoing LCMV protocol, so we would like to refrain from this experiment at the current stage.

Instead, we decided to compare DCs from control and *XCR1.Bcl6^{KO}* in an in vitro cross-presentation assay, which allowed us to normalize the cDC1 input number. The data is now added as new Suppl. Fig. 4 and supports the notion that mLN cDC1 retain the ability to cross-present antigen to CD8 T cells.

- For OVA immunization, the authors used ovalbumin grade V (OVA, Sigma Aldrich) and poly(I:C) (Sigma-Aldrich) that likely contain endotoxin.

We thank the reviewer for this comment. This will likely contribute to a more potent immune response and making our findings more generalizable. Nevertheless, in our newly added experiments in Fig. 8 and Suppl. Fig 8 we now used NP-OVA (for assessment of OT-II cells on day 7) and NP-KLH (for assessment of endogenous Tfh and Tfr cells on day 14 after immunization as well as for the assessment of total vs. high-affinity NP-specific antibody responses) conjugates from Biosearch Technologies that should contain much less endotoxin, if any, than the aforementioned substances from Sigma-Aldrich.

Reviewer #2 (Remarks to the Author):

This study by Xiao et al. performed comprehensive analysis of three distinct conditional Bcl6-deficient mouse models for conventional dendritic cell phenotype, transcriptome and function. The authors showed that deletion of Bcl6 altered the phenotype and transcriptome of cDC1 and cDC2, but their lineage identity is preserved. Functionally, the authors found the Bcl6-deficient mice maintained their ability to cross-present antigen to CD8+ T cells, but induced fewer T follicular helper cells in response to the model antigen ovalbumin and mounted diminished TH17 immunity to *Citrobacter rodentium* in the colon. While this study is overall well performed and presented, two major concerns need to be addressed before it can be considered for publication.

We thank the reviewer for the positive comments on our manuscript and for the insightful suggestions. The biggest concerns were the gating that we used to characterize cDCs, given that CD11c was previously shown to be downregulated in the absence of Bcl6, potentially biasing our data. To address these concerns, we added all relevant gating strategies as supplementary data, showing that we used very broad gates to assure the complete capture of cDCs. More importantly, we added data making use of the Clec9A-reporter by gating on YFP instead of CD11c, showing that the outcome of Bcl6 deficiency remains the same, no matter the gating strategy. We believe that these added controls strengthen the paper significantly. Specific improvements were made as follows:

1. The biggest concern about this study is the claim of “Bcl6 deficiency diminished cDC1 in the periphery, and Notch2-dependent cDC2 in the spleen and intestinal lamina propria”. As the authors have discussed in their introduction, a previous study showed that “down-regulation of CD11c in the absence of Bcl6 may have compromised the detection of” cDCs. In current study, the authors still used CD11c to define classical DCs, it is unclear whether Bcl6 deficiency diminished periphery cDC1 and Notch2-dependent cDC2, or Bcl6 deficiency only down-regulated CD11c in those cDC subsets. An alternative method that is independent of CD11c expression is needed to draw this important conclusion, for example using *Zbtb46* expression to define classical DCs. More specifically:

1) In Figure 1, for splenic and resident LN cDC analysis, could the authors show their gating strategy for cDCs?

We have now included the complete gating strategies for spleen, pLN, mLN and SILP as a new Supplementary Figure 1.

2) In Figure 2, for migratory LN cDC and small intestine lamina propria cDC analysis, could the authors show their gating strategy for cDCs? An alternative method for defining cDC besides pre-gate on CD11c is needed here to distinguish cDC deficiency with CD11c down-regulation.

CD11c-gated organs are now included in the newly added gating strategy in the new Supplementary Figure 1. We further added a side-by-side comparison of gating strategies using CD11c versus CLEC9A reporter activity on spleen DCs as new Supplementary Figure 6.

We have also performed DC subset staining on mLN and pLNs in *CLEC9A.Bcl6^{KO}* and control mice and added those data to this rebuttal (see below). As described in the manuscript, resident and migratory DCs are differently affected by the lack of Bcl6, meaning that resident and migratory DCs need to be analyzed separately. We are not aware of any established ways to differentiate those two populations without the use of CD11c, so we included a CD11c vs MHCII gate downstream of the YFP gate (see figure). We paid specific attention to not exclude cells based on low CD11c expression and essentially all YFP⁺ cells are included in either the resident or migratory population. Since we are unable to report DC numbers without the use of CD11c in lymph nodes, and we only performed this experiment once with suboptimal compensation, we decided not to include these data in the manuscript. We however show them here as they show again that the phenotype caused by Bcl6-deficiency is not an artifact caused by downregulation of CD11c.

3) In Figure 6, for panels A and C, could the authors explain in the figure legend the pre-gating strategy of the splenocytes, or whether the splenocytes underwent DC enrichment? Why is there only CD11c positive cells? As the authors discussed in their result section, “the ESAMlo subset of cDC2 expressed slightly lower levels of CD11c compared to their ESAMhi counterparts”. Due to the difference in CD11c level between

ESAMhi and ESAMlo cDC2 subsets, an alternative method for defining cDC besides pre-gate on CD11c is required to draw the conclusion that “Bcl6 deficiency in DC causes the loss of the ESAMhi lineage of splenic cDC2”.

We missed reporting that the cells in Figure 6 A and C were pre-gated on CD11c and thank the reviewer for catching that. To address the valid concern of a lack of alternative gating strategies, we have now replaced the original Figure 6E by one that depicts pre-gating on the YFP-reporter only (dropping the CD11c pre-gate). The added YFP-based gating strategy using *Clec9a.Bcl6^{KO}* on the YFP reporter (new suppl. Fig 6) further supports this point.

4) In Figure 7C, the authors still used CD11c as the pan-DC marker, loss of CD11c positive cells in microscopy can be due to the loss of DC or down-regulation of CD11c expression. An alternative DC marker is required to draw the conclusion of “confirmed a substantial loss of DC from this region in the CD11c.Bcl6KO spleen”.

We have now added new microscopy images in which we made use of the YFP reporter under the control of *Clec9a* (Figure 8B). The YFP signal is much brighter than the CD11c signal, but both signals overlap. Importantly, the YFP signal is equally lost from the marginal zone bridging channels supporting the notion that DCs are absent from this location when lacking Bcl6.

2. The authors discovered a surprising finding that XCR1+ cDC that preserved in Bcl6-deficient mice express the cDC2 marker CD11b. It would be important to determine whether this reflects acquisition of CD11b by cDC1, or cDC2 with aberrant XCR1 expression. But the strategy chosen by the authors raises some concern. To answer this question, could the authors sort XCR1+CD11b- cDC, XCR1+CD11b+ cDC and XCR1-CD11b+ cDC separately from CD11c.Bcl6KO mice, and compare their transcriptional profile with XCR1+CD11b- cDC and XCR1-CD11b+ cDC from WT mice. The XCR1 positivity should be determined by XCR1 level in WT mice. Principal component analysis of these datasets would answer the above question in a more unbiased manner. With current data, it is not clear whether the XCR1+ cDC (independent of CD11b expression) are cDC1 with aberrant CD11b expression, or are a combination of cDC1 with a third XCR1+CD11b+ cDC population.

We realized that the original plots in Fig. 1A in the paper used unfortunate scaling of the of the data and have now adjusted the plots to the way the data is presented in the rest of the paper. The new version of the plot shows better that XCR1⁺CD11b⁻ cDCs are absent in the transgenic mice, and indeed all XCR1⁺ cells uniformly express CD11b. Given the PCA clustering shown in Fig. 3A and prominent alignment of Bcl6-deficient XCR1⁺CD11b⁺ cDCs with wt cDC1 as seen in the heatmap in Fig. 3B, we are confident that those cells are indeed cDC1 that upregulated CD11b, and not cDC2 that upregulated XCR1. We do not have single-cell RNA-seq data and feel that PCA clustering with classical cDC markers would be heavily biased. As an alternative, we chose to add cDC2 data from control mice to the histograms in Fig. 4A as dotted lines, which serves as a useful reference and shows that XCR1⁺CD11b⁺ cDC broadly maintain a cDC1 phenotype. Importantly, those histograms also show unimodal distribution of cDC1 markers by the XCR1⁺CD11b⁺ population, again arguing that the bulk sequencing data is not biased towards a remaining cDC1 population within a XCR⁺CD11b⁺ non-cDC1 pool. Together, we are confident that XCR1-expressing DCs in Bcl6-deficient models are indeed cDC1.

Minor point:

In Figure 8C, could the two panels of colon length be mis-labelled?

In our hands, the effect on colon length is indeed stronger at day 9, and we assume that healing is well underway by day 21 despite the continuous presence of elevated numbers of T cells.

Reviewer #3 (Remarks to the Author):

The manuscript by Hongkui Xiao et al. titled "Loss of Bcl6 transcriptionally alters classical dendritic cells and diminishes T follicular helper cell and Th17-inducing cDC2 in mice" describes Bcl6-deficiency in the DC modifies the phenotype as well as the gene expression of cDC1 and cDC2. There are controversial data regarding the role of TF Bcl6 in all cDC, and these findings point to a relevant impact of Bcl6-deficiency in the immunological function of cDC1 and cDC2. The authors have done an exciting study addressing the impact of Bcl6 deletion in murine cDC, but additional comments and concerns are raised below.

We thank the reviewer for the enthusiasm concerning our study. Below are the responses to the remaining concerns. We believe that added data based on the suggestions improve the manuscript.

Major points.

Figure 1 and Figure 2. The absence of Bcl6 results in exciting findings that cCD1 can acquire some cDC2 markers. The same double-positive cells were observed in the LNs and SI lamina propria. Since the XCR1+CD11b+ cDC are so evident in different compartments, it would be interesting to include a specific flow cytometry analysis in XCR1/CD11b double positive cells, delimiting the gate only in XCR1+/CD11b+ (figure 1 A). Also, it will be very informative to include the frequency (%) inside of the gate and also a graph showing the MFI of CD11b comparing all groups. Indeed, the expression of CD11b (MFI) in CD11c.Bcl6Ko is clearly more pronounced in comparison to XCR1.Bcl6Ko, and in the periphery and mesenteric resident LN most of the DCs of XCR1+ shift to the CD11b+ population.

We realized that the original plots in Fig. 1A in the paper used unfortunate scaling of the of the data and have now adjusted the plots to the way the data is presented in the rest of the paper. The new version of the plot shows better that XCR1+CD11b- cDCs are absent in the transgenic mice, and instead all XCR1+ cells uniformly express CD11b. Importantly, the histograms in Fig. 4A also point to a unimodal distribution of all markers by the XCR1+CD11b+ population, again arguing that the entire XCR1+ cDC population upregulates CD11b in the absence of Bcl6.

We have added the % to Fig. 1A and apologize for missing that earlier.

We have now also added MFI data on CD11b across the different genotypes for all organs analyzed. The data is added as new Supplementary Figure 3. This again shows that the scaling in the original Fig. 1 was unfortunate, and that CD11b is equally upregulated in cDC1 in the two mouse models, while it is only upregulated in cDC2 in the CD11c-driven cre-model, as these cells are not targeted in the XCR1-driven deletion model. Interestingly, the already high CD11b levels in cDC2 are further upregulated in the absence

of Bcl6 specifically in systemic DCs (spleen and resident LN populations), while those in peripheral DCs (migratory LN populations, SILP and lung) seem to remain at wildtype levels. This is an intriguing finding further supporting environmental aspects of Bcl6 regulation. We thank the reviewer for the suggestion of depicting this.

The figure 4B shows that the absence of Bcl6 modulates some relevant gene ontology pathways that could modulate some immunological functions of XCR1+ cDC, for example, the production of IL-1, regulation of cell adhesion, and response to the virus. Have you considered whether the XCR1/CD11b double-positive cells acquire gain from one of these immunological functions? For example, splenic DCs can secrete substantial amounts of mature IL-1 β upon stimulation with TLR ligands, such as TLR3, that were upregulated on splenic XCR1.Bcl6KO. Could XCR1/CD11b double-positive cells express more Nlrp3 and produce a higher amount of IL-1 β ? Even though the results from Figure 6 show interesting findings that the absence of Bcl6 does not affect the capacity to cross-present antigen, it would be interesting to evaluate if so; other immunological functions could be affected, such as in virus infection or in the presence of TLR ligands.

Further analysis of gene expression datasets revealed the IL-6 pathway as potentially deregulated in Bcl6-deficient cDC1. This analysis is now added as new Supplementary Figure 5. To test whether this had in vivo relevance, we injected *XCR1.Bcl6^{KO}* mice intraperitoneally with poly(I:C), which efficiently targets TLR3, highly and uniformly expressed in cDC1 from control and *XCR1.Bcl6^{KO}* mice (see Figure 4A). Indeed, *XCR1.Bcl6^{KO}* mice show higher serum levels of IL-6 two hours post injection, suggesting that Bcl6-deficient cDC1, directly or indirectly, contribute to higher systemic IL-6 levels upon activation. These data are now added to the manuscript as new Figure 5D. To improve the flow of the manuscript, we changed the order of the cross-presentation panel and the GO-term analysis, first showing data that supports the claim that Bcl6-deficient XCR1+ DCs are indeed cDC1 (Fig. 4), and then showing the impact that Bcl6-deficiency has on these cells (Fig. 5).

Pg7, line 197. The author described, "OTI proliferation was also observed in the mLN. In the mLN of XCR1.Bcl6KO mice OTI cells showed somewhat diminished proliferation compared to control mice; however, this most likely reflects the decreased number of migratory cDC1 in the mLN of XCR1.Bcl6KO mice (Fig. 2A)." I agree that the impairment of migration cDC observed previously could explain the lower proliferation of OT-I. However, to confirm if these cells preserve the ability to cross-present antigen to OT-I, instead of transferring OT-I into recipients, it would be interesting to sort cDC1 from XCR1.Bcl6KO mice were previously immunized with heat-shocked OVA-MEFs and co-culture in vitro with OT-I cells to evaluate cell proliferation.

We agree with the reviewer that this would be a nice experiment, which unfortunately proves to be technically very challenging. Cross-presentation by migratory cDC1 from the intestines has previously been addressed using genetic models expressing ovalbumin in the intestinal epithelium (Joeris et al., Science Immunology 2021) but crossing these mice to our *XCR1.Bcl6^{KO}* would require several generations, unfortunately exceeding our timelines. Difficulties in in vivo loading of peripheral cDC1 with injected dead cells together with the scarcity of migratory cDC1 in Bcl6-deficient mice made us fail in generating meaningful data using this approach.

Instead, we decided to perform the suggested experiment with the difference of pre-loading enriched DCs with OVA *ex vivo* followed by extensive washing before co-culture with OT-I cells. The data is now added as new Suppl. Fig. 4 and supports the notion that mLN cDC1 retain the ability to cross-present antigen to CD8 T cells.

Pag. 8, line 210. The authors described, "The remaining cDC in the spleen of CD11c.Bcl6KO mice also showed somewhat lower expression of MHCII and higher expression of CD11b than control cDC2 (Fig. 6A)." As observed in figure 6A, splenic cDc from CD11c.Bcl6KO mice express higher CD11b than control; however, the percentage of MHC-II on these cells was similar in both groups, control (96.4%) and CD11c.Bcl6KO (92.6%). It seems like the MFI of MHC-II is lower in CD11c.Bcl6KO than the control. However, there needs to be more information about the MFI in this figure. I suggest including the data points graph pooled from independent experiments in addition to the representative dot and contour plots.

We have now added mean MFI values and average deviation of MHCII in Fig. 6A. The data is pooled from 3 different experiments with 3 mice each.

Figure 8. The deficiency of Bcl6 affects the Th17 priming at day 9; however, after 21 days, most parameters were recovered (colon length, IL-17+IL-22; IL-17+/IFN-g; IFN-g+IL-17- cells). Even though the Th17 differentiation was affected initially, in the discussion, the authors described, "We cannot currently explain the differences in the early innate immune response between CD11c.Notch 2KO and CD11c.Bcl6KO mice, but our data are consistent with the idea that DC play the dominant role in coordinating the adaptive immunity required to terminate the infection. However, while Bcl6-dependent cDC2 support optimal Th17 induction to *C. rodentium*, they are not essential for this response, as Th17 numbers normalized over time." Since we observed a slight increase in the IL-17+ L-22; IL-17+/IFN-g cell number after 9 days, have you considered that Th17 cell expansion in infected CD11c.Bcl6ko mice, could explain the cell numbers normalized over time (after 21 days)?

We agree with the reviewer that this is a very likely scenario. Our leading hypothesis is that in *CD11c.Bcl6^{KO}* mice, fewer Th17 cells are generated early after infection due to a deficiency in optimal priming. This leads to a prolonged infection, in line with delayed healing. Th17 numbers however recover over time, putatively due to continuous expansion in the presence of *Citrobacter*, ultimately clearing the infection. We have added this thought to the discussion (page 17, line 466).

Minor points

Pg4, line 110. Be consistent in describing XCR1-CD103+CD11b+ population. In Figure 2, the graph was indicated as XCR1-CD11b+ CD103+.

We changed the text accordingly.

REVIEWER COMMENTS

Reviewer #1 (Remarks to the Author):

In the revised manuscript titled "Loss of Bcl6 transcriptionally alters classical dendritic cells and diminishes T follicular helper cell and Th17-inducing cDC2 in mice", the authors provided additional data on Tfr and antibody responses as I suggested previously. The new data, however, showed that Bcl6 in DCs influences Tfr, not in Tfh.

The clec9a^{cre}Bcl6^{fl/fl} mice had similar CXCR5^{hi}PD1^{hi} Tfh cells as the Bcl6^{fl/fl} mice (Figure 8G, 8H). There were also litter differences in antibody response, especially the high-affinity anti-NP2 antibody in the immunized clec9a^{cre}Bcl6^{fl/fl} mice (Figure 8J). In the text, the authors stated that "the frequencies of GC Tfh cells, gated as CXCR5^{hi}PD-1^{hi} or CXCR5^{hi}Bcl6^{hi}, were not altered in Clec9a.Bcl6KO mice at day 14 after immunization (Fig. 8G, H). In contrast, we observed a strong reduction in CXCR5^{hi}PD-1^{hi} or CXCR5^{hi}Bcl6^{hi} Tfr cells among Foxp3⁺ CD4⁺ T cells (Fig. 8G, H), indicating that Bcl6 in DCs is particularly important for the generation of Tfr cells". (Line 301-304). This is an accurate interpretation of the data.

However, in the abstract, the authors still stated that "DC-targeted Bcl6-deficient mice induced fewer T follicular helper..... in response to immunization" (line 43~44). The similar statement remains in the title "Loss of Bcl6 transcriptionally alters classical dendritic cells and diminishes T follicular helper cell.." Such statements are misleading.

The current manuscript represents a large body of work. However, the authors need to follow the data and change their interpretation accordingly.

Reviewer #2 (Remarks to the Author):

The authors put a lot of effort in the revision process, but their claim "Bcl6 deficiency diminished cDC1 in the periphery" is still not well supported. To support their claim, I have suggested: An alternative method that is independent of CD11c expression is needed to draw this important conclusion, for example using Zbtb46 expression to define classical DCs. More specifically: 2) In Figure 2, for migratory LN cDC and small intestine lamina propria cDC analysis, could the authors show their gating strategy for cDCs? An alternative method for defining cDC besides pre-gate on CD11c is needed here to distinguish cDC deficiency with CD11c down-regulation.

In the response letter, the authors replied:

CD11c-gated organs are now included in the newly added gating strategy in the new Supplementary Figure 1.

Thanks to the authors for providing the complete gating strategy. In the current panel for peripheral LNs, the authors have not excluded Langerhans cells, and they will highly contaminate the migratory cDC gate in the peripheral LNs.

We further added a side-by-side comparison of gating strategies using CD11c versus CLEC9A reporter activity on spleen DCs as new Supplementary Figure 6.

Thanks for the side-by-side comparison between CD11c versus CLEC9A reporter analysis in spleen DC. But this cannot help to demonstrate the deficiency of cDC1 in periphery (migratory cDC in peripheral LNs, mesenteric LNs and small intestine lamina propria).

We have also performed DC subset staining on mLN and pLNs in CLEC9A.Bcl6KO and control mice and added those data to this rebuttal (see below). As described in the manuscript, resident and migratory DCs are differently affected by the lack of Bcl6, meaning that resident and migratory DCs need to be analyzed separately. We are not aware of any established ways to differentiate those two populations without the use of CD11c, so we included a CD11c vs MHCII gate downstream of the YFP gate (see figure). We paid specific attention to not exclude cells based on

low CD11c expression and essentially all YFP+ cells are included in either the resident or migratory population. Since we are unable to report DC numbers without the use of CD11c in lymph nodes, and we only performed this experiment once with suboptimal compensation, we decided not to include these data in the manuscript. We however show them here as they show again that the phenotype caused by Bcl6-deficiency is not an artifact caused by downregulation of CD11c. Thanks to the authors for their great effort. But unfortunately, the Clec9A-Cre can only label subsets of DC, thus can only provide limited support to their conclusion. More specifically, in the original paper (ref 36), Clec9A-Cre traced 90% of cDC1, but only 40% of cDC2 in migratory cDC from peripheral LNs (ref36 Fig. 3D). When using CD11c in the new Sup Fig. 1, the authors show a ratio of migratory to resident cDC as 84:8 in mLN, while using Clec9a in the response letter show a ratio of migratory to resident cDC as 52:38. Similar results are also found in pLN analysis. This difference indicates the Clec9a-Cre lineage tracing is not a good tool for identifying cDC. In addition, in both Fig. 2 and this new figure in the response letter, XCR1+ cDC number varied dramatically among individual biological replicates in the control group. There is always less than 50% of the mice show a difference when compared to Bcl6-deficient mice. Due to the above reasons, I find the conclusion "Bcl6 deficiency diminished cDC1 in the periphery" not well supported by the current data.

Reviewer #3 (Remarks to the Author):

The authors have performed most of the suggested experiments and adequately responded to the concerns raised by the reviewers. The results of additional experiments confirmed the significance of the mechanism described in this manuscript. In its current form, the study represents a significant contribution to the field, and the manuscript has been significantly improved. I have no additional concerns.

Point-by-point response to the reviewer's comments on NCOMMS-22-50737-B

We thank the reviewers for their supportive comments and hope that we were able to fully address remaining concerns with this revised version and the additional data shown below.

Reviewer #1 (Remarks to the Author):

In the revised manuscript titled "Loss of Bcl6 transcriptionally alters classical dendritic cells and diminishes T follicular helper cell and Th17-inducing cDC2 in mice", the authors provided additional data on Tfr and antibody responses as I suggested previously. The new data, however, showed that Bcl6 in DCs influences Tfr, not in Tfh.

The *Clec9a*^{cre}*Bcl6*^{fl/fl} mice had similar CXCR5^{hi}PD1^{hi} Tfh cells as the *Bcl6*^{fl/fl} mice (Figure 8G, 8H). There were also litter differences in antibody response, especially the high-affinity anti-NP2 antibody in the immunized *Clec9a*^{cre}*Bcl6*^{fl/fl} mice (Figure 8J). In the text, the authors stated that "the frequencies of GC Tfh cells, gated as CXCR5^{hi}PD-1^{hi} or CXCR5^{hi}Bcl6^{hi}, were not altered in *Clec9a*.*Bcl6*^{KO} mice at day 14 after immunization (Fig. 8G, H). In contrast, we observed a strong reduction in CXCR5^{hi}PD-1^{hi} or CXCR5^{hi}Bcl6^{hi} Tfr cells among Foxp3⁺ CD4⁺ T cells (Fig. 8G, H), indicating that Bcl6 in DCs is particularly important for the generation of Tfr cells". (Line 301-304). This is an accurate interpretation of the data.

However, in the abstract, the authors still stated that "DC-targeted Bcl6-deficient mice induced fewer T follicular helper..... in response to immunization" (line 43~44). The similar statement remains in the title "Loss of Bcl6 transcriptionally alters classical dendritic cells and diminishes T follicular helper cell.." Such statements are misleading.

The current manuscript represents a large body of work. However, the authors need to follow the data and change their interpretation accordingly.

Response:

We thank the reviewer for acknowledging the large body of work, including the newly added experiments as suggested by the reviewer. We agree that DC-specific *Bcl6* deficiency had the biggest effect on Tfr cells on day +14 (presented in Fig. 8G,H). However, Bcl6-deficiency also resulted in reduced frequencies of early CXCR5⁺Bcl6⁺ Tfh OT-II cells after OVA + poly(I:C) i.p. immunization in *CD11c*.*Bcl6*^{KO} mice (Suppl. Fig. 8A) and reduced frequencies of CXCR5⁺PD-1⁺ and CXCR5⁺Bcl6⁺ Tfh OT-II cells recovered from day +7 *Clec9a*.*Bcl6*^{KO} hosts following NP-OVA/alum i.p. immunizations (Fig. 8D and Suppl. Fig. 8B). Tfh cell differentiation of OT-II cells, which themselves cannot differentiate into Tfr cells, was therefore clearly affected by DC-specific Bcl6-deficiency. Furthermore, while the endogenous GC Tfh cells, gated as CXCR5^{hi}PD-1^{hi} or CXCR5^{hi}Bcl6^{hi}, were not altered in *Clec9a*.*Bcl6*^{KO} mice at day 14 after immunization (Fig. 8G, H, and Suppl. Fig 8C), as pointed out by the reviewer, frequencies of endogenous CXCR5^{int}PD-1^{int} and CXCR5^{int}Bcl6^{int} Tfh cells were also reduced (Fig. 8G, H, and Suppl. Fig 8C). Together with the somewhat impaired antibody responses, these data indicate that Bcl6 in DCs is responsible for proper Tfh and Tfr cell differentiation. We would like to point out that we had discussed the implications of reduced Tfr and unchanged GC Tfh cells responses on day 14 in the discussion as follows: "Given the stronger reduction in Tfh cells in *Clec9a*.*Bcl6*^{KO} mice at early stages following immunization, as compared to later stages, it is possible that a continued lack of Tfr-mediated suppression of Tfh cells may allow for a compensatory outgrowth of Tfh cells over time, thus mitigating the net effects of DC-specific Bcl6 deficiency. This would be in line with the multistep differentiation process of Tfh cells in which B cells become the sole antigen-presenting cell subset in GCs,

with no classical DCs being present in GCs⁵⁵.” To accommodate for the reviewer’s comments, we have revised the title, abstract, and the relevant parts of the results and discussion sections where appropriate to emphasize that Bcl6-deficiency impacted Tfr cells but also Tfh cells to some extent.

Reviewer #2 (Remarks to the Author):

The authors put a lot of effort in the revision process, but their claim “Bcl6 deficiency diminished cDC1 in the periphery” is still not well supported. To support their claim, I have suggested: An alternative method that is independent of CD11c expression is needed to draw this important conclusion, for example using Zbtb46 expression to define classical DCs. More specifically: 2) In Figure 2, for migratory LN cDC and small intestine lamina propria cDC analysis, could the authors show their gating strategy for cDCs? An alternative method for defining cDC besides pre-gate on CD11c is needed here to distinguish cDC deficiency with CD11c down-regulation. In the response letter, the authors replied: CD11c-gated organs are now included in the newly added gating strategy in the new Supplementary Figure 1.

Thanks to the authors for providing the complete gating strategy. In the current panel for peripheral LNs, the authors have not excluded Langerhans cells, and they will highly contaminate the migratory cDC gate in the peripheral LNs.

We further added a side-by-side comparison of gating strategies using CD11c versus CLEC9A reporter activity on spleen DCs as new Supplementary Figure 6.

Thanks for the side-by-side comparison between CD11c versus CLEC9A reporter analysis in spleen DC. But this cannot help to demonstrate the deficiency of cDC1 in periphery (migratory cDC in peripheral LNs, mesenteric LNs and small intestine lamina propria).

We have also performed DC subset staining on mLN and pLNs in CLEC9A.Bcl6KO and control mice and added those data to this rebuttal (see below). As described in the manuscript, resident and migratory DCs are differently affected by the lack of Bcl6, meaning that resident and migratory DCs need to be analyzed separately. We are not aware of any established ways to differentiate those two populations without the use of CD11c, so we included a CD11c vs MHCII gate downstream of the YFP gate (see figure). We paid specific attention to not exclude cells based on low CD11c expression and essentially all YFP+ cells are included in either the resident or migratory population. Since we are unable to report DC numbers without the use of CD11c in lymph nodes, and we only performed this experiment once with suboptimal compensation, we decided not to include these data in the manuscript. We however show them here as they show again that the phenotype caused by Bcl6-deficiency is not an artifact caused by downregulation of CD11c.

Thanks to the authors for their great effort. But unfortunately, the Clec9A-Cre can only label subsets of DC, thus can only provide limited support to their conclusion. More specifically, in the original paper (ref 36), Clec9A-Cre traced 90% of cDC1, but only 40% of cDC2 in migratory cDC from peripheral LNs (ref36 Fig. 3D). When using CD11c in the new Sup Fig. 1, the authors show a ratio of migratory to resident cDC as 84:8 in mLN, while using Clec9a in the response letter show a ratio of migratory to resident cDC as 52:38. Similar results are also found in pLN analysis. This difference indicates the Clec9a-Cre lineage tracing is not a good tool for identifying cDC. In addition, in both Fig. 2 and this new figure in the response letter, XCR1+ cDC

number varied dramatically among individual biological replicates in the control group. There is always less than 50% of the mice show a difference when compared to Bcl6-deficient mice.

Due to the above reasons, I find the conclusion “Bcl6 deficiency diminished cDC1 in the periphery” not well supported by the current data.

We thank the reviewer for the recognition of our efforts and the continuous input leading to additional important controls. The remaining concern of the reviewer is that peripheral cDC1 may not be diminished. We agree with the reviewer that Clec9A is not an ideal driver to target all DCs, as peripheral cDC2 are not targeted efficiently. Since we report absolute numbers for cDC1s, which are efficiently targeted in the Clec9A model, we do not believe this to be a problem, especially since most data is based on the analysis of mice lacking Bcl6 in XCR1.cre mice.

We agree that the DC recovery in our pLN samples is generally quite variable. Nevertheless, cDC1 are diminished consistently and significantly across all peripheral cDC1 subsets in XCR1.Bcl6^{KO} mice (mLN, SILP, lung, and pLN). The defects in the migratory pLN cDC1 populations are indeed a little hard to appreciate, and below are plots depicting the same data as shown in main Figure 1B and 2A on a logarithmic axis (left) and as a ratio of migratory versus resident cDC1 in the pLN (right) (panel A in the figure below). The impact of Bcl6 on migratory cDC1 numbers is easier to see in this visualization.

The reviewer was further concerned that our pLN cDC1 population was contaminated by Langerhans cells. Langerhans cells do not express XCR1, so we do not expect any contamination across the cDC1 populations reported across in the manuscript. Nevertheless, we decided to perform the staining previously suggested by the reviewer, in which we added antibodies for zbtb46 and EpCAM (see Figure below, panel B). To allow for gating on migratory cDC1 in the absence of CD11c staining, we added an antibody against CD103 and used this to gate on cDC1. As published by others before (Wu, ..., Murphy; JEM 2016), Langerhans cells express quite high levels of zbtb46. If gated on the EpCAM^{neg} population, the phenotype of a deficiency in migratory cDC1 in the absence of Bcl6 remains. Together, we show in different organs using different staining panels that migratory cDC1 are diminished in numbers if lacking Bcl6.

A pLN Data from Figure 1B and 2A

B New pLN stain using zbtb46 and CD103

Reviewer #3 (Remarks to the Author):

The authors have performed most of the suggested experiments and adequately responded to the concerns raised by the reviewers. The results of additional experiments confirmed the significance of the mechanism described in this manuscript. In its current form, the study represents a significant contribution to the field, and the manuscript has been significantly improved. I have no additional concerns.

We thank the reviewer for the supportive comments.

REVIEWERS' COMMENTS

Reviewer #1 (Remarks to the Author):

The authors have addressed my comments with a modified title and Abstract. I am satisfied.

Reviewer #2 (Remarks to the Author):

The authors have made an admirable effort to address the issues raised in the previous review rounds. I have no additional concerns.

Point-by-point response to the reviewer's comments on NCOMMS-22-50737-C

Reviewer #1 (Remarks to the Author):

The authors have addressed my comments with a modified title and Abstract. I am satisfied.

Reviewer #2 (Remarks to the Author):

The authors have made an admirable effort to address the issues raised in the previous review rounds. I have no additional concerns.

We thank the reviewers for their constructive feedback and fair comments, which have improved our manuscript.